# Streaming Detection of Queried Event Start

**Cristóbal Eyzaguirre**
Stanford University

**Eric Tang**
Stanford University

**Shyamal Buch**
Stanford University

**Adrien Gaidon**
Stanford University

**Jiajun Wu**
Stanford University

**Juan Carlos Niebles**
Stanford University

## Abstract

Robotics, autonomous driving, augmented reality, and many embodied computer vision applications must quickly react to user-defined events unfolding in real time. We address this setting by proposing a novel task for multimodal video understanding—Streaming Detection of Queried Event Start (SDQES). The goal of SDQES is to identify the beginning of a complex event as described by a natural language query, with high accuracy and low latency. We introduce a new benchmark based on the Ego4D dataset, as well as new task-specific metrics to study streaming multimodal detection of diverse events in an egocentric video setting. Inspired by parameter-efficient fine-tuning methods in NLP and for video tasks, we propose adapter-based baselines that enable image-to-video transfer learning, allowing for efficient online video modeling. We evaluate four vision-language backbones and three adapter architectures in both short-clip and untrimmed video settings.

## 1 Introduction

The ubiquity of embodied vision applications, such as robotics [1], autonomous driving [2], and augmented reality [3], highlights the need for methods that can *detect the occurrence of events with low latency in untrimmed and egocentric video streams*. Despite significant strides made in video understanding, a review of the existing literature reveals a notable gap: most current methods are designed for batch processing or adopt windowed approaches that result in redundant computation when new frames are considered. These approaches are effective in addressing existing benchmark tasks, but they fall short in practical, real-time applications due to the high computational overhead required in processing additional new frames and their limited context.

Some prior work has attempted to bridge this gap by extending traditional offline action recognition [4, 5, 6] and detection [7, 8] tasks to an online setting [7, 9]. In particular, online detection of action start (ODAS) [9, 10] emphasizes low-latency[1] detection of when an action from a predefined list of classes begins in a streaming video starts. ODAS captures the urgency of the detection task, but it cannot assess settings where the user wants to specify more complex event queries beyond the scope of the predefined classes, nor do existing benchmarks for ODAS focus on egocentric settings, which are important for embodied applications. Consequently, approaches trained on existing datasets for online action detection may be constrained by the limited range of events they are designed to recognize, reducing their applicability to more diverse or unforeseen scenarios.

---

[1]Total latency is the sum of two factors: (1) computation latency, the time taken by a model to run on hardware and generate a prediction for a given frame, and (2) observation latency, the number of frames of the event that the model has to see before it identifies the specified query event has started to occur in the video. In this paper, we propose new metrics that consider *observation latency*, while also reporting existing metrics of model efficiency.

38th Conference on Neural Information Processing Systems (NeurIPS 2024) Track on Datasets and Benchmarks.

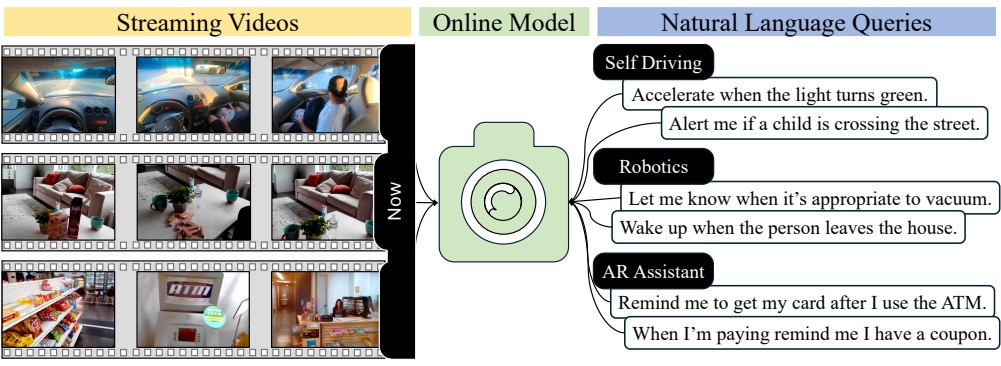

Figure 1: **Overview of our proposed SDQES task**. The goal of streaming detection of queried event start (SDQES) is for a system to detect the *start* of a complex event, described by *natural language*, with low latency from a *streaming* video input. This task is a novel intersection of multimodal event and online/streaming video understanding benchmarks. It is intended to encourage the design of new streaming multimodal models for challenging *egocentric* or embodied settings (*e.g.*, assistive robotics, augmented reality) where time-sensitivity is a key concern for safety, accessibility, or convenience.

Natural language, in contrast, enables users to flexibly specify complex events. However, traditional tasks for this kind of multimodal setting, such as temporal localization with language [11, 3], are typically offline, requiring full observation of the complex event (and potentially, of the video as a whole [12, 13]) before providing the output detection. Thus, models for this task are not suitable for deployment in online settings where time sensitivity is paramount, such as those shown in Figure 1. This gap underscores the need for new methods that enable low-latency, real-time detection of complex events specified through natural language in untrimmed and egocentric video streams, as well as datasets to train and benchmark on.

To this end, we propose a novel task at the unique intersection of online and multimodal video understanding: *Streaming Detection of Queried Event Start* (SDQES). The goal of SDQES is to detect the start of a complex event, described by a natural language query or description, with high accuracy and low latency, with a particular focus on egocentric video streams in embodied applications. To support this, we present a new benchmark, EgoSDQES, leveraging annotations from the comprehensive Ego4D dataset [3]. This task synthesizes three significant challenges. First, it operates under a streaming framework where models access only past video frames without future data, such that the model will need to use precursor visual cues to provide timely detection outputs. Second, SDQES demands a multimodal approach, incorporating language queries that require a profound understanding of the video's content, which means that models cannot rely on a small set of cues to distinguish a closed vocabulary of atomic actions. Lastly, the task also aims to enable progress on applications with egocentric video inputs, which often involve complex issues like variable camera angles and motion blur that effective streaming systems must learn to address.

We propose baseline methods that extend existing vision language foundation models by adding adapters to enable online applications with real-time outputs on untrimmed videos with constant time per additional frame. This approach leverages the pretraining of vision language foundation models for parameter-efficient adaptation and transfer to the task of event detection in video streams. By adapting foundation models into an efficient streaming video architecture, we combine the strengths of massively diverse vision pretraining sets with the specific requirements of real-time video processing.

In sum, we make three contributions. First, we formulate Streaming Detection of Queried Event Start (SDQES), a novel task for online multimodal video understanding representing a unique intersection of challenges for event detection models. Second, we construct a new benchmark based on the existing large-scale egocentric video dataset. We propose metrics suited for measuring progress on this streaming multimodal task. Third, we propose a mechanism to adapt existing pretrained vision foundation models to handle long streaming videos efficiently. We evaluate multiple combinations of vision backbones and adapter architectures on both short clips and extremely long videos.

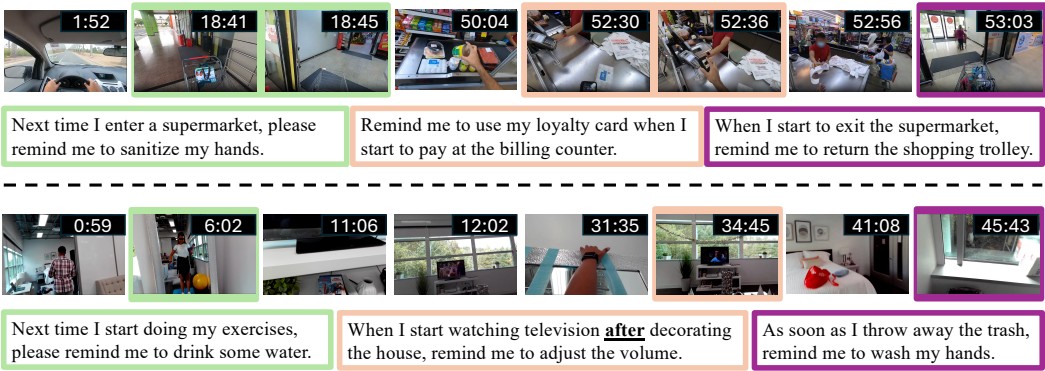

Figure 2: **Example videos and queries** from our dataset EgoSDQES.

## 2 Related Work

Our proposed streaming detection of queried event start (SDQES) task is a unique intersection of video understanding areas that have not been explored by prior work, which we summarize below.

**Action Recognition and Detection in Videos.** The goal of action recognition systems is to output the action or activity present in an input video clip [14, 15]. While in recent years the focus has broadened to settings with untrimmed videos, these tasks are often designed with a fixed vocabulary of action classes [16, 17], and models have been primarily developed for offline usage where the model has access to the full video before outputting predictions [18, 19, 20, 21, 4]. This limits their immediate efficacy in online or streaming contexts, or in settings where a more flexible model is required to handle open-vocabulary complex event descriptions.

**Online Detection of Action Start (ODAS).** To address the limitations of offline tasks, Online Detection of Action Start was introduced [9], which built upon prior work in the field of early action recognition [22, 23, 24] and online action anticipation [7, 25]. Given an untrimmed video stream input, the goal of ODAS [9, 10] is to detect the *start* of an action, from a set of pre-defined action classes [17, 16]. This is intended to also be useful in settings where having low total latency is paramount. On the other hand, in our proposed task of SDQES, the events are described by open-vocabulary natural language, an increased challenge for video understanding models that require the design of different technical approaches and appropriate evaluation protocols.

**Action Anticipation.** Like ODAS and SDQES, action anticipation [26, 27, 28, 29, 30, 31] involves the timely prediction of events in a video sequence from a predetermined set of action classes. However, unlike traditional ODAS, action anticipation predicts the action class for a frame in the future instead of focusing on the present frame. These models typically do not predict a "background class," which is a distinguishing feature of SDQES and ODAS. Most of them are traditionally offline, processing pre-recorded video data rather than streaming video, whereas SDQES is designed for online, real-time detection. As in ODAS, the output of action anticipation is a classification to a predetermined set of classes, rather than the start of an event specified in language. To our knowledge, no prior work has combined natural language event specification with online prediction.

**Video Understanding with Language.** There is significant work on video event understanding with language across datasets, models, and tasks [32, 33, 34, 35, 36, 37, 38, 39]. The most related to our work is the task of event localization with language [40, 34, 11, 13], where the goal is to take a language query and untrimmed video as input and to provide the localization of the queried event in the video. Others have considered anticipating which of two possible events is more likely given videos and additional dialogue transcriptions [41], with connections to commonsense reasoning [42]. However, these are offline tasks or use auxiliary information to make predictions over a limited event space. Our work inherits the complexity of using natural language queries for events, while simultaneously extending to the challenging area of online/streaming video understanding.

**Egocentric Video Understanding.** Egocentric video data poses unique challenges relative to traditional video data often explored in video action understanding. Prior work [3, 1, 43, 44, 45], aiming to capture settings representative for assistive robotics, driving, and augmented reality, has

allowed access to rich collections of challenging egocentric visual data. Previous tasks in this domain have focused mainly on traditional event understanding, such as offline action recognition [43] or localization with language [3], so models developed for these tasks [12, 46, 47, 48, 49] are not directly suitable for online or streaming detection settings. Egocentric action anticipation [44, 30], has been constrained to specific atomic action vocabularies, which limits their effectiveness in settings where users may need to specify more complex events flexibly. We aim to address these limitations for the egocentric setting.

**Adaptation of Pretrained Vision Models for Video.** Due to the computational expense of fully fine-tuning video models, a recent line of work inspired by the use of adapter modules in NLP [50, 51] has focused on parameter-efficient fine-tuning of image models, in particular the CLIP vision encoder [52], for offline video tasks [53, 54, 55, 56]. Notably, ST-Adapter [54] and AIM [56] achieve strong results in action recognition by learning spatio-temporal adapter modules over a CLIP backbone. More recently, another line of work has focused on two-channel models that utilize intermediate CLIP features in order to avoid backpropagation through frozen ViT parameters [57, 58, 59]. Our proposed model builds upon these paradigms for efficiently transferring representations for video understanding, and further extends them for effectively handling the online video setting.

## 3 SDQES Task: Formulation and Metrics

### 3.1 Formulation and Goal

Given an input video stream $V$ and an event query in natural language $q$, the goal of SDQES is to accurately provide the temporal location where the described event starts with low latency. Let $V_{\text{stream}}^{(i)} = \{f_1, f_2, \ldots f_i\}$ be a streaming input video sequence of frames up to the current frame $f_i$ at time $i$, and $t_s$ the start time of the queried event in the video stream. A model $M$ for SDQES is

$$M(V_{\text{stream}}, q) \mapsto t_{\text{out}}, \tag{1}$$

where the goal is to output a *high accuracy* prediction of event start (*i.e.*, output time $t_{\text{out}} = t_s$) with low latency. Since the ground truth start time $t_s$ is not known to the model in advance and inputs are processed sequentially, we do not restrict the model to a single prediction. Instead, the model may output a set of prior predictions $t_{\text{out}} < t_s$. Thus, an additional goal is one of *high precision*, where such false positive outputs by model $M$ are minimal.

### 3.2 Metrics: Accuracy with Low Latency

**Existing Protocols.** The seminal prior work in early detection, MMED [22], reported a comprehensive evaluation protocol including FPR, accuracy, and timeliness metrics. However, these do not provide a complete picture of the actual model performance for SDQES. Specifically, the FPR and accuracy metrics are measured at the frame level and may not be representative of the model's performance on the actual task. Additionally, the metric for timeliness assumes frame-perfect annotations, where, in practice, there can be reasonable disagreement about when an event starts. Later work in Online Detection of Action Start (ODAS) [19, 9] instead adopts a single evaluation metric: p-mAP. This metric addresses the noise in annotations by considering action starts as correct when they are contained within a temporal window. However, p-mAP disregards temporal order; thus, it is not strictly online. More discussion in the supplementary material.

**Streaming Recall.** Our key accuracy metric is *streaming recall* of event start. SR builds on p-mAP and extends the definition to account for the greater ambiguity present in SDQES by considering the *first $k$ predictions*. Following from Eq. 1, let $P_M^{(k)} = \{t_{\text{out}_1}, \ldots, t_{\text{out}_k}\}$ denote a set of the first (up to) $k$ predictions $t_{\text{out}}$ generated by a model $M(V_{\text{stream}}, q)$, and let $t_s$ denote the groundtruth start time of the event described by query $q$. A model output set $P_M^{(k)}$ is then "correct" if and only if

$$\exists t_{\text{out}}' \in P_M^{(k)} : -anticipation \leq t_s - t_{\text{out}}' \leq latency, \tag{2}$$

where $anticipation$ and $latency$ define the asymmetric temporal tolerance window. Since $k$ represents the *first* predictions in the set, Eq. 2 penalizes models with a high false positive rate, as such models would exhaust their $k$ guesses of the event start early in the stream. By considering different values of $k$ and tolerances, we can provide fine-grained measurements of a model's capabilities.

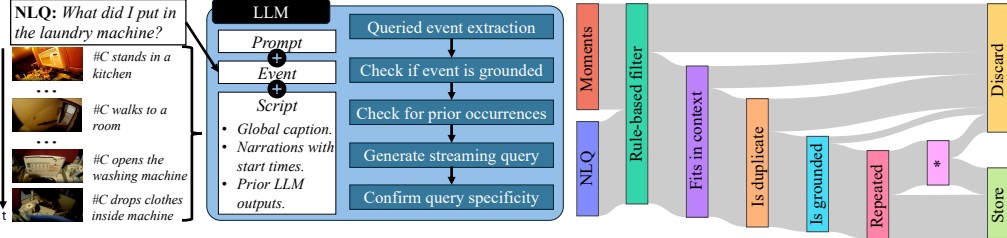

(a) Overview of data annotation pipeline.

(b) Sankey Diagram.

Figure 3: **Dataset generation pipeline**. Left: we show the generation pipeline steps for an example video with dense captions. Right: Sankey diagram illustrates the flow of data from Ego4D through the various filtering stages. *Asterisk* ($*$) encodes a filter based on query specificity.

**Streaming Minimum Distance.** In addition to Streaming Recall, we propose a second metric that focuses on timeliness: *Streaming Minimum Distance* (SMD). This metric measures the average error of the closest prediction made by the model $M$ to the groundtruth $t_s$. Given the set of predictions $P_M$, we define the minimum distance as $d_{\min} = \min_{t_{\text{out}} \in P_M} |t_s - t_{\text{out}}|$.

We report this metric as SMD@$k$, which measures the average minimum distance across all queries with groundtruth start times $t_s$ and across a model's first $k$ predictions $t_{\text{out}}$ This metric is complementary to the streaming recall metric, providing a measure of the temporal accuracy of the model's predictions. More details of both metrics are provided in the supplementary material.

**Model Efficiency.** In addition to the proposed metrics that focus on model task performance and *observational latency*, we evaluate the computational efficiency of our models to assess their suitability for real-time applications across a range of metrics that reflect both computational resource usage and response speed. Our computational efficiency metrics include Parameter Count for assessing memory footprint, Multiply-Accumulate Operations (MACs) and Floating Point Operations (FLOPs) to quantify computational load, and Computation Latency to measure model processing time per frame.

# 4   Data Collection and Annotation

As our task is novel, no dataset has been previously created for it. Instead, we repurpose publicly available datasets with temporally grounded language annotations. Specifically, we focus on Ego4D [3], a recent large-scale dataset of long videos from an egocentric perspective. This dataset is challenging because it contains diverse activities, viewpoints, and camera motion, making it ideal for evaluating the robustness of our method to challenging real-world scenarios. Furthermore, it is easily accessible under the Ego4D license. Additionally, we demonstrate how the dataset can be extended with other video sources by applying the same pipeline to the videos and annotations from the EgoExoLearn dataset [60]. Details specific to this other dataset are provided in the supplementary material.

Our innovative data generation pipeline employs Large Language Models (LLMs) for generation and several key filtering steps, with stages illustrated in Fig. 3a and 3b. Figure 3a illustrates the process of how the LLM contributes to modifying a single event, and generates relevant metadata for subsequent filtering. Figure 3b shows the flow of the existing temporal annotations through the successive filtering stages, some of which use outputs from the LLM. We describe each of these below.

**Generation Pipeline.** Specifically, we start with Ego4D's temporally grounded *Moments* (Action Localization) and *NLQ* (Natural Language Queries) annotations, as well as dense video captions (*Narrations*). For each annotation, the LLM extracts the event, as this is the key information needed. For instance, from the query "Where did I last leave the box?", we extract the event "leave box." It then confirms the event's reflection in narrations to ensure contextual accuracy and groundedness in the video content, a necessary step since the queries must refer to visible events.

Next, the LLM refers to the narrations to verify if the extracted event has previously occurred in the video. This prior check ensures that previous occurrences can be accurately identified, avoiding misclassification due to missing narrations. The LLM is then prompted to generate an original

| Dataset | Task | Video Source | View | #Videos | #Annotations | Video duration (s) |
|---|---|---|---|---|---|---|
| Thumos 14 | ODAS | YouTube | Allocentric | 413 | 6365 | <180 |
| ActivityNet | ODAS | YouTube | Allocentric | 15K | 22.6K | <180 |
| NLQ | Temporal localization | Ego4D | Egocentric | 1046 | 17052 | 492 |
| Moments | Temporal localization | Ego4D | Egocentric | 1189 | 19151 | 472 |
| EgoSchema | Video QA | Ego4D | Egocentric | 5063 | 5063 | 180 |
| EgoSDQES | SDQES | Ego4D | Egocentric | 1773 | 12767 | 1553 |

Table 1: **Comparison of various related datasets.**

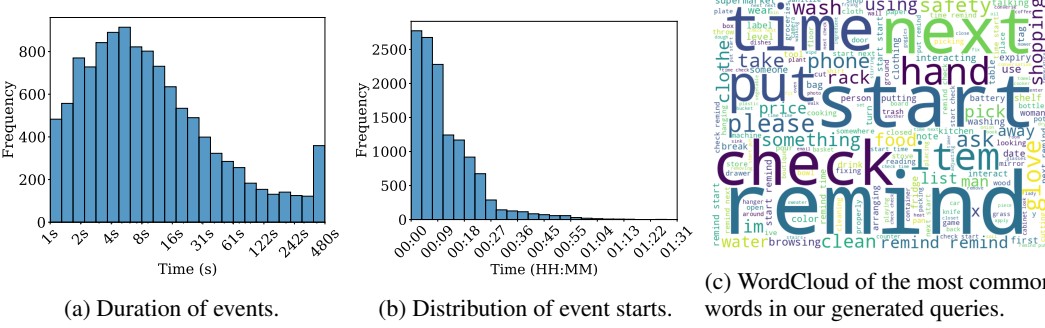

(a) Duration of events.  (b) Distribution of event starts.  (c) WordCloud of the most common words in our generated queries.

Figure 4: **Dataset statistics**. Left: Event duration in seconds. Center: Distribution of Event Start with respect to video start. Right: Word Cloud of the query generations.

streaming query for the current instance, following the template of setting a reminder to do something when the event begins. We prompt the model to disambiguate so that the query cannot refer to another prior event if one was detected. Finally, we verify the specificity of the generated query to the annotated event instance using the LLM to differentiate between multiple instances of the same event.

The crucial filtering stage ensures data quality and relevance. We eliminate annotations based on fixed rules using metadata to quickly discard obviously irrelevant annotations. We discard annotations that cannot be verified within the LLM's context window (8k tokens). This is necessary to avoid truncation, which would compromise the LLM's capacity to identify if the event has occurred before.

Finally, there is a bifurcation in the filtering process. For events occurring for the first time, we use the generated annotation as is, ensuring that new events are accurately captured and added to the dataset. If the event has occurred before, we add a final filter stage to check if the generated query is unambiguous using the LLM. This specificity check is key because the dataset is intended for detecting events in a streaming video, where future video content is unknown. Thus, we ensure that queries are specific and identifiable without relying on context from future portions of the video.

**EgoSDQES.** We run the full pipeline using GPT-4 [61] as the LLM, resulting in 12,767 annotations for over 740 hours of video. We split the videos and annotations following the original Ego4D train/val split, resulting in 1,331 training and 442 validation videos. These videos are untrimmed and contain at least one streaming query. Table 1 compares our dataset with other egocentric video datasets. Our dataset is larger than NLQ, Moments, and EgoSchema in terms of both the number of videos and annotations. Notably, our videos have a much longer duration (1,553 seconds on average) compared to other datasets, even those used for ODAS, which are limited to 180 seconds.

Note that because of the filter that discards annotations with scripts that do not fit in the context (and pre-existing biases in Ego4D annotations), our final dataset mostly contains queries that refer to events in the first 30 minutes of a video (see Figure 4b). Nevertheless, our dataset videos are substantially longer, so our annotations occur further into the videos compared to existing datasets.

Figure 4c shows the most common words in the generated queries, illustrating their diversity. Common words such as "next," "time," "remind," and "start" reflect the nature of the task, which involves setting reminders for future events or actions related to common objects and situations.

Datasheets, data cards, and visualizations of the annotations are in the supplementary material.

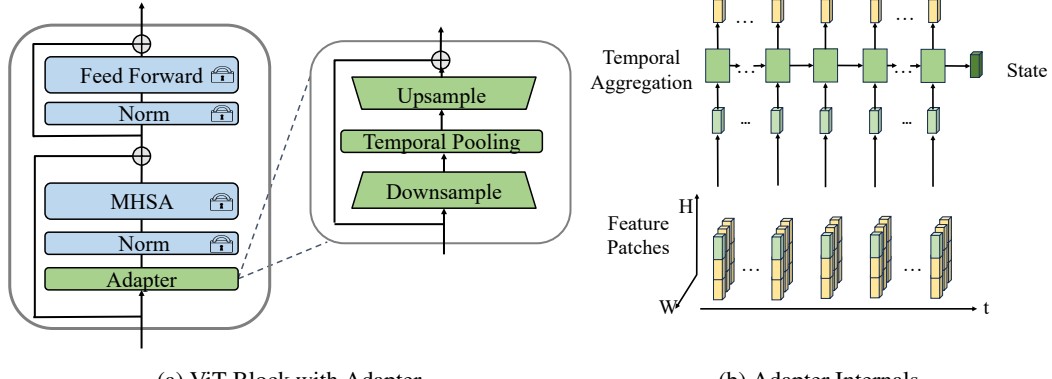

(a) ViT Block with Adapter                    (b) Adapter Internals

Figure 5: **Overview of the Streaming-Adapter.** (a) Intervened Block: the lock icon denotes frozen parameters - only adapter parameters are trained. Temporal adapters operate on a reduced dimension for efficiency. (b) Adapter Internals: the adapter operates over the temporal dimension and consists of temporal aggregation layers. The final state of the model is stored for when the next frame arrives.

## 5  Baseline Approaches

We present adapter-based baseline models that enable efficient adaptation of vision-language models for streaming video input, allowing for online video modeling. We provide an overview of their model architectures and training objectives. The implementation details are in the supplementary material. Our models serve as initial explorations into this new problem setting, laying the groundwork for future work to build upon. Figure 5 shows an overview of our architecture.

### 5.1  Streaming-Adapters

Given a pre-trained image model $\mathcal{F}$ and a set of videos, our objective is to bridge the modality gap between the image-level model pretraining and the spatio-temporal video task. We adapt the image model into a spatio-temporal video model $\mathcal{F}*$, while reusing as many parameters from $\mathcal{F}$ as possible.

A common strategy for both adapting image models to video processing tasks [54, 56, 55, 53], as well as for developing video models from the ground up [62, 63], involves incorporating temporal aggregation layers into the pre-existing ViT architecture (Figure 5a). This enables the model to reason over successive video frames by adding new temporal-specific layers between the ViT's spatial layers for temporal aggregation across image patches.

A key consideration is the choice of architecture for the temporal aggregation layer, as this fundamentally governs how the model's computational requirements scale with the input sequence length (*i.e.*, video duration). For streaming video-language tasks, it is desirable to use architectures that can efficiently process new frames with a constant computational cost, rather than requiring re-computation over the entire sequence. To this end, we explore different architectural choices suitable for streaming, such as recurrent models and 1D convolutional models, which can incrementally update their representations as new frames arrive. Specifically, we use them to aggregate temporal information across timesteps for each patch-wise tubelet with shape, as shown in Figure 5b.

Importantly, the adapters operate in a reduced dimension $d' < d$ to reduce computational cost. Also note that architectures that use convolutions require zero-padding to produce an equal-sized output. We pad the sequence on the left to ensure that no information from future frames is leaked into the past. Additional baseline architecture details are provided in the supplementary material.

### 5.2  Training and Loss

**Data Sampling.** One key issue for detecting event starts, as also observed in prior work on ODAS [9], is the significant imbalance between positive and negative samples in the training data. As videos are long and events are infrequent, there are many more frames that are not event starts than those that are. To mitigate this impact, we reformulate the training approach by leveraging denser supervision

signals provided by a dense labeling task. In this task, we pair sampled windows of $w_s$ frames $f_{i-w_s+1}, \ldots, f_i$ (along with query $q$) with ground truth labels $y_{i-w_s+1}, \ldots, y_i$, where each label $y$ is true if the associated frame belongs to the region of the video corresponding to query $q$. For further details, please refer to the supplementary material.

**Loss Formulation.** Our model employs a cross-entropy loss function, which operates over the cosine similarities between the frame embeddings $\mathbf{e}_{f_i}$ obtained from the video encoder and the query embedding $\mathbf{e}_q$ from the corresponding language encoder. The goal is to maximize the similarity between the frame embeddings that correspond to the specific event described by the query and the query embedding itself. For a set of frame embeddings $\{\mathbf{e}_{f_1}, \ldots, \mathbf{e}_{f_N}\}$ and the query embedding $\mathbf{e}_q$, the cosine similarity $s_i$ is given by $s_i = \frac{\mathbf{e}_{f_i} \cdot \mathbf{e}_q}{\|\mathbf{e}_{f_i}\| \|\mathbf{e}_q\|}$. We then apply the sigmoid function to these similarities to model the probability $p_i$ of each frame embedding corresponding to the query. The binary cross-entropy loss $\mathcal{L}$ is computed as $\mathcal{L} = -\sum_{i=1}^{N} y_i \log(p_i)$, where $y_i$ is the ground truth label indicating whether frame $f_i$ is relevant to the query $q$. Finally, because events typically occur for only a fraction of the video (see Figure 4a), there are many more negative frames than positive ones. Therefore, we also apply a weighting scheme to the loss function during training.

# 6 Experiments

We evaluate a variety of combinations of Streaming Adapters and dual-encoder vision-language models, including the current state-of-the-art (SOTA) egocentric video encoder.

In no particular order, we consider adapters based on: 1) 1D Convolutions (which we refer to as ST-Adapter as they closely resemble the adapter in [54]); 2) Quasi-Recurrent Neural Networks [64], a more computationally efficient gated RNN (referred to as QR-Adapter); and 3) RetNet [65], a close analog to the standard Transformer architecture [66] that allows for low-cost inference by linearizing the attention mechanism (RN-Adapter in our experiments). Additionally, we also consider a standard non-temporal MLP adapter, refered to simply as Adapter in the experiments.

In this study, we evaluate several vision-language backbone models, including CLIP as detailed in the work by Radford et al. [52]. Additionally, we extend our evaluation to incorporate vision-language models that have been pretrained using egocentric video data. It is important to highlight that our focus is primarily on dual-encoder models. This design choice is based on the efficiency consideration that modeling the query and the video with a dual-encoder does not require reprocessing the video for each new query. Among the models assessed, we include the well-known EgoVLP [12] and LaViLA [46], and the current state-of-the-art dual encoder model EgoVideo [47].

## 6.1 Experimental Setup

We initialize backbone weights to the best-performing pretrained models available and then freeze them. For EgoVLP, LaViLa, and EgoVideo, we modify each architecture to process a single frame at a time, diverging from their original multi-frame input configurations. Unless otherwise specified, we use the *Base* variants of all encoders. Adapters are added at the beginning of each block and before the MLP layers, with weights initialized to approximate an identity operation at the start of training. Dimensions are selected such that all adapters have roughly the same amount of hyperparameters.

All models are trained on 60-frame windows sampled at 1 frame per second (FPS), except for RetNet-based adapters, which use 30 frames to ensure stability, and models based on the EgoVideo backbone, which are limited to 30 frames due to memory constraints. We predict action starts by thresholding the cosine similarities between frame embeddings and the query embedding, with the Streaming Recall metric's anticipation and latency set to 5 and 10 seconds, respectively. Additional details are available in the supplementary material.

**Efficiency and Latency Measurement.** To assess model efficiency, we include metrics for both modified single-frame backbones (e.g., EgoVLP backbone) and unmodified versions using a sliding window of four frames. We measure computation latency by running each model on a full video and recording the total elapsed time, with results averaged over three runs to ensure consistency. Latency values reflect hardware and implementation specifics and may vary under different conditions, such as with different accelerators or environments.

| Method | 1 Min. | | 5 Min. | | | | | |
| --- | --- | --- | --- | --- | --- | --- | --- | --- |
| | SR@1↑ | SMD@1↓ | SR@1↑ | SR@2↑ | SR@3↑ | SMD@1↓ | SMD@2↓ | SMD@3↓ |
| Zero-Shot CLIP | 16.9 | 24.3 | 7.9 | 11.6 | 14.0 | 151.3 | 140.3 | 132.6 |
| CLIP + Adapter | 19.5 | 23.5 | 8.9 | 13.7 | 17.2 | 135.7 | 121.7 | 113.3 |
| CLIP + QR-Adapter | 23.7 | 21.2 | 9.1 | 14.1 | 18.7 | 136.7 | 117.7 | 102.9 |
| LaViLa + Adapter | 19.5 | 23.4 | 8.7 | 13.0 | 16.2 | 163.4 | 151.7 | 144.0 |
| LaViLa + QR-Adapter | 29.1 | 18.1 | 9.3 | 12.8 | 16.5 | 132.1 | 115.9 | 104.1 |
| EgoVLP + Adapter | 18.1 | 24.0 | 8.4 | 13.0 | 16.7 | 160.8 | 148.7 | 141.5 |
| EgoVLP + QR-Adapter | 28.8 | 17.7 | 9.7 | 14.1 | 17.9 | 133.1 | 120.8 | 110.9 |
| EgoVLP + ST-Adapter | 17.4 | 30.5 | 8.6 | 13.4 | 17.0 | 170.7 | 161.4 | 155.6 |
| EgoVLP + RN-Adapter | 25.7 | 21.3 | 9.4 | 15.4 | 20.1 | 174.8 | 159.0 | 149.2 |
| EgoVideo + Adapter | 27.1 | 28.8 | 16.0 | 21.8 | 26.4 | 148.5 | 138.3 | 131.2 |

Table 2: **Baseline Results** for CLIP [52], EgoVLP [12], and LaViLa [46] fine tuned with a variety of adapters.

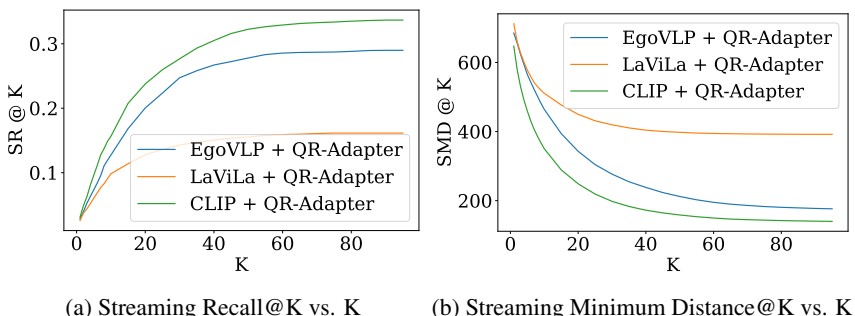

(a) Streaming Recall@K vs. K    (b) Streaming Minimum Distance@K vs. K

Figure 6: **Full length video results**.

## 6.2 Task Performance Results

In our experimental evaluation, we present findings on the performance of various adapter models integrated with dual-encoder vision-language architectures, using video clips of 1-minute and 5-minute duration captured at 1 frame per sec. Table 2 summarizes these findings.

**Impact of New Dataset on Task Performance.** Our results demonstrate a clear improvement in model performance when trained with our data. We include a zero-shot single-frame baseline that uses CLIP [52] to illustrate the capabilities of the dual-encoder without additional training. Across all tested backbones, every adapter model outperformed the zero-shot CLIP baseline. This improvement underscores the effectiveness of training on our generated dataset.

**Temporal Adaptation with QR-Adapter.** We find that our QR-Adapter-based model consistently outperforms the zero-shot baseline and the standard non-temporal Adapter across all backbones. This supports our hypothesis that temporal adaptation, as introduced by QR-Adapter, is beneficial for SDQES.

**Alternative Streaming Temporal Adapters.** We further this analysis by including additional formulations for temporal streaming adapters based on alternative architectures. We find that, while the 1D convolution-based ST-Adapter showed limited success, the linear-attention-based RN-Adapter showed comparable performance to the QR-Adapter, supporting the claim that more complex temporal modeling capabilities are required for SDQES.

**Extension to Untrimmed Video Results.** We also assess model performance on full-length videos up to two hours long: Figure 6a shows the relationship between the number of predictions allowed (modulated by the K-value) and the mean Streaming Recall. This trade-off between recall and the volume of predictions is particularly relevant for longer videos, where the chance of capturing relevant events increases with more predictions. Figure 6b details the precision of predictions, showing that higher K-values are more likely to include predictions that are closer to the ground truth annotations.

| | Memory | Computational Latency | | |
|---|---|---|---|---|
| Model | # parameters | Multiply Adds | Floating Point Operations | Latency |
| EgoVLP backbone | 180.92 M | 7.85 TMACs | 15.7 Tflops | 1.68 s |
| EgoVLP + Adapter | +7.9% | +12.7% | +12.8% | +15.5% |
| EgoVLP + ST Adapter | +7.9% | +12.7% | +12.8% | +18.5% |
| EgoVLP + QRNN Adapter | +7.5% | +12.0% | +12.2% | +21.5% |
| EgoVLP + RetNet Adapter | +7.6% | +15.2% | +15.3% | +99.5% |
| EgoVLP Sliding Window | +0.1% | +298.5% | +298.8% | +260.2% |

Table 3: **Model Efficiency.** This table compares the number of model parameters along with the computational cost of processing a single frame for each listed architecture. For computational latency we report both the total number of operations (Multiply-Accumulates operations and floating point operations) along with the processing time taken on a single V100 graphics processor to run on a 5 minute and 50 seconds long video taken from the dataset: $video\_uid = dd08bc58 - b614 - 4ba7 - b883 - a213560621dd$.

## 6.3 Latency and Model Efficiency

Latency is critical for real-time applications such as assistive technologies, human-computer interaction, and autonomous systems. Our models are designed for efficiency, featuring minimal parameters, low FLOPs, and reduced latency to meet these demands. Table 3 presents the memory and computation requirements of different adapter models. The context-less backbone architecture is the most memory-efficient and fastest option. However, all our adapters are highly efficient, introducing only about a 13% increase in operations compared to the backbone alone. Notably, our temporal adapters—ST-Adapter, QRNN-Adapter, and RetNet-Adapter—approach the efficiency of the non-temporal vanilla adapter.

In contrast, the Sliding Window variant consumes four times the computational resources and is limited to only four seconds of context. Regarding latency, all temporal adapters, except for the RetNet-Adapter, exhibit computation times comparable to the non-temporal adapter, adding approximately 5% to the total computation time. The higher latency of the RetNet-Adapter is due to the absence of an optimized CUDA implementation, whereas the QRNN-based adapters benefit from a dedicated CUDA kernel, and the ST-Adapter leverages PyTorch's efficient convolution operations. These results demonstrate that our proposed models effectively balance performance and efficiency, making them suitable for real-time applications requiring both rapid response and accurate temporal modeling.

## 7 Conclusion

We have introduced Streaming Detection of Queried Event Start (SDQES), a novel task designed to push the boundaries of online multimodal video understanding, with a specific focus on the challenges presented by egocentric video streams. The task synthesizes the unique complexities of detecting complex events in a streaming framework, requiring both high accuracy and low latency. Our contributions include the formulation of SDQES, which demands that the models operate without future data, relying instead on past video frames and language cues to predict events as they unfold. We have also developed a benchmark, leveraging existing egocentric videos and annotations, and proposed metrics tailored for evaluating progress in this streaming setting. We propose adapter-based baseline approaches to serve as a starting point. Our temporal adapter models highlight the benefits of incorporating temporal adaptation, as introduced by QR-Adapter, for this task. Importantly, these models achieve enhanced performance while maintaining low latency, making them suitable for real-time applications where rapid response is essential.

**Limitations.** Our dataset inherits any errors or omissions present in the Ego4D narrations that were used to generate it. Additionally, the narrations may lack important details, violating assumptions about their quality. As is the case for ODAS, the inherent ambiguity in defining precise start and end points of actions remains a challenge.

## Acknowledgments and Disclosure of Funding

This work was in part supported by the Toyota Research Institute (TRI) and ONR N00014-23-1-2355.

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

# A  Official Checklist

1. Submission introducing **new datasets** must include the following in the supplementary materials:

   (a) Dataset documentation and intended uses. Recommended documentation frameworks include datasheets for datasets, dataset nutrition labels, data statements for NLP, and accountability frameworks.
      - See Section G for complete datasheet.

   (b) URL to website/platform where the dataset/benchmark can be viewed and downloaded by the reviewers.
      - Website is available at sdqesdataset.github.io.

   (c) URL to Croissant metadata record documenting the dataset/benchmark available for viewing and downloading by the reviewers.
      - Croissant metadata is available at sdqesdataset.github.io/dataset/croissant_metadata.json.

   (d) Author statement that they bear all responsibility in case of violation of rights, etc., and confirmation of the data license.
      - We, the authors, bear all responsibility in case of violation of rights or any other legal issues arising from the use of this dataset. We also confirm that the data license is as follows: The dataset is published under the MIT License. However, Ego4D videos are licensed under a separate Ego4D License as cited in [3].

   (e) Hosting, licensing, and maintenance plan. The choice of hosting platform is yours, as long as you ensure access to the data (possibly through a curated interface) and will provide the necessary maintenance.
      - The data is hosted in the open Github repository associated with the website github.com/sdqesdataset/sdqesdataset.github.io/. The repository will be maintained by the authors.

2. To ensure accessibility, the supplementary materials for **datasets** must include the following:

   (a) Links to access the dataset and its metadata.
      - Both data and metadata are hosted in the Github repository associated with the website github.com/sdqesdataset/sdqesdataset.github.io/.

   (b) The dataset itself should ideally use an open and widely used data format. Provide a detailed explanation on how the dataset can be read. For simulation environments, use existing frameworks or explain how they can be used.
      - The main dataset deliverable is the CSV file that contains the generated queries. This file can be found here. Each row includes details such as the dataset split ('train', 'val'), the source ('moments', 'nlq'), unique identifiers for the video (video_uid) and the clip (clip_uid), and the annotator (annotator_uid). It also includes an annotation index (ann_idx, referring to the index of the annotation withing the specific annotators annotations), the generated query, the corresponding response, and additional metadata from the generation process. Additionally, it provides the start and end times of the event within the video (in seconds), the video's frames per second (video_fps), and its total length (video_length).

   (c) Long-term preservation: It must be clear that the dataset will be available for a long time, either by uploading to a data repository or by explaining how the authors themselves will ensure this.
      - The dataset has been uploaded to GitHub and is publicly accessible at https://sdqesdataset.github.io/dataset/all.csv.

   (d) Explicit license: Authors must choose a license, ideally a CC license for datasets, or an open source license for code (e.g. RL environments).
      - EgoSDQES is published under MIT License. Note that Ego4D videos licensed under a separate Ego4D License [3]. We additionally open source the code for dataset generation github.com/sdqesdataset/sdqes_generation.

   (e) Add structured metadata to a dataset's meta-data page using Web standards (like schema.org and DCAT): This allows it to be discovered and organized by anyone. If you use an existing data repository, this is often done automatically.

- We host the website, data, and associated repositories using https://github.com.

(f) Highly recommended: a persistent dereferenceable identifier (e.g. a DOI minted by a data repository or a prefix on identifiers.org) for datasets, or a code repository (e.g. GitHub, GitLab,...) for code. If this is not possible or useful, please explain why.

- We host the data on github sdqesdataset.github.io/dataset/all.csv. Furthermore, we include a sha256 hash of the dataset in the Croissant metadata file.

3. **For benchmarks**, the supplementary materials must ensure that all results are easily reproducible. Where possible, use a reproducibility framework such as the ML reproducibility checklist, or otherwise guarantee that all results can be easily reproduced, i.e. all necessary datasets, code, and evaluation procedures must be accessible and documented.

- Code for all baselines, metrics and dataloaders is included at github.com/sdqesdataset/sdqes_baselines. Additionally, Sections 6 and D include most of the implementation details for every baseline. Finally, Table 2 includes a Weights and Biases link (wandb.ai) to the specific experiment hyperparameters, metrics, and logs for every experiment in the main paper.

# B  Discussion of Metrics

## B.1  Human Baseline

As part of our ongoing effort to establish a human baseline for comparison, we evaluate human expert performance in the untrimmed video setting. Due to the length and associated costs of video annotation by humans, we focus on a subset of 87 annotations. Our results show that $SR@1[-5, 10]$ is 72.4, and $SR@3[-5, 10]$ is 86.2. This serves as a clear indicator of the high quality of our data, as evidenced by robust human performance. Furthermore, it substantially surpasses the best model performance in the same setting (see Figure 6a). We will continue to verify more annotations, but the current results already indicate more than reasonable generation quality, and human performance remains clearly superior.

## B.2  Comparison to Prior Protocols

Section 3.2 of this work introduced a new metric for evaluating online detection of queried event start (SDQES), specifically designed to deal with challenging ambiguities in Streaming Detection of Queried Event Start. Here, we expand on our prior discussion. In particular, we reiterate that our setting naturally entails additional ambiguities compared to both ODAS [9] and TALL [11]. The streaming setting, where low-latency constraints mean models must make predictions with a higher degree of contextual ambiguity than offline models, means that there is a high potential for false positive outputs. Evaluation metrics for SDQES therefore should account for false positive rates.

The seminal prior work in early detection, MMED [22], reported a comprehensive evaluation protocol that accounted for the False Positive Rate (FPR), while also including two other metrics for measuring the timeliness of the predictions, and accuracy. However, each metric represents an incomplete picture of the actual model performance. Specifically, the metrics for FPR and accuracy are measured at the frame level and therefore aren't necessarily representative of the model's performance on the actual task of predicting action starts in the context of full video streams. Furthermore, the metric for action start timeliness assumes frame-perfect annotations as an action start made one frame before the annotated region doesn't count towards the metric. In practice, there can be reasonable disagreement (e.g. within 1 second) of when an event formally starts.

Because of the limitations listed above later work in Online Detection of Action Start (ODAS) [9] have moved away from the protocols in [22] and instead use a single evaluation metric: **Point-Level AS Detection mAP** (p-mAP). The metric addresses the noise in annotations by considering action starts as correct when they are contained within a temporal window centered around the annotated action start. Additionally, the same window is adjusted to measure timeliness, as window widths represent the maximum accepted latency in the action start detection. In brief, a tighter window is met only by predictions with a small temporal offset to the annotated start. Critically, the metric has a key limitation: its handling of temporal order. Sorting event detections inherently ignores the temporal sequence of these detections, making the metric temporally invariant. This is illustrated in

*Q: Let me know when you see the paper towels.*

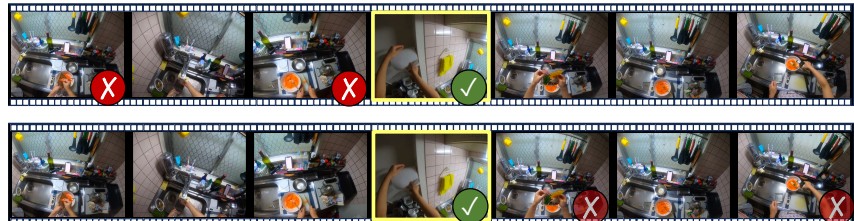

Figure 7: **Metric Comparison.** These two scenarios are *indistinguishable* for the p-mAP metric. Top: model correctly alerts the user on the third attempt after two false positives. Bottom: model is successful on its first attempt and alert can be turned off, avoiding two future false positives.

Figure 7. In a streaming context, it's essential to consider the temporal flow of events, as it directly affects the real-time decision-making process.

The metric proposed in this paper, **Streaming Recall** (SR), builds on Point-Level AS Detection mAP, but extends the definition to account for the greater ambiguity present in SDQES. Like p-mAP predicted action starts are considered correct when the temporal offset to the annotated start is lower than a specified maximum accepted latency, *i.e.* when the prediction is within a window. The main differences are: First, we consider $K$ predicted actions starts to account for the increased ambiguity in our setting. Second, to make the metric more representative of the streaming setting we consider action starts temporally, selecting the *first K*. That is, we favor early predicted action starts and suppress later ones (anything that comes later after the set "limit" of $K$ outputs is hit). This contrasts with p-mAP where predictions are suppressed based on model confidence, instead of by temporal order alone.

By construction, our metric implicitly accounts for the *temporal false positive rate* in the action start predictions as models with high false positive rate will quickly exceed the $k$ predictions limit. We find that considering multiple predictions is especially important for long videos like those present in the *Untrimmed* setting for Ego4D [17].

Furthermore, like p-mAP, our Streaming Recall metric also uses windows which allow it to quantify timeliness while accounting for noise in the annotations. As a more direct measure of timeliness we also introduce *Streaming Minimum Distance at K*. This other metric tracks the offset between the best out of the *first K* predicted action starts and the annotated one. As in case of Streaming Recall, we find that considering more predictions improves the model's measured timeliness. This again indicates that many false positive starts are predicted before finding the right one.

Additionally, we note that our proposed metrics are not limited to a single instance of the event, even if our dataset only provides annotations for a single instance.

## B.3 Choice of Window Width

In addressing the issue of event start time ambiguity mentioned in Limitations 7, we incorporate a temporal tolerance into our Streaming Recall metric during evaluation. This adjustment helps minimize the impact of this ambiguity by allowing us to account for predictions that are *close enough* to the ground truth annotations, even if they are not exact.

Following the nomenclature in Section 3.2, we adopt a data-centric approach to determine the appropriate values for *anticipation* and *latency* based on this consideration. We utilize the fact that multiple annotators may label the same event in Ego4D to identify overlaps, referred to as 'collisions,' in *Moments* annotations. These collisions are detected by verifying whether any two annotated events achieve an Intersection over Union (IoU) score of at least 0.7 and share identical high-level semantic descriptors. From these identified pairs, we calculate the average variance in event start times across the training data, which amounts to $\sigma^2 = 28.8$ seconds. The square root of this variance, 5.3 seconds, represents the standard deviation of the event start times.

We therefore set the *anticipation* and *latency* values in the Streaming Recall metric to be respectively $\sigma$ and $2 * \sigma$, which, when discretized into our framerate of 1 FPS results in a tolerance window of $[-5, +10]$ seconds. While arbitrary, we choose to set a larger value for the allowed *latency* than for the allowed *anticipation* for two reasons. First, when deduplicating annotations in the generation

| Model | $SR@1[5, 10]$ | $SR@1[2, 5]$ |
|---|---|---|
| ZS CLIP ViT-B/16 | 16.90% | 8.85% |
| CLIP Vanilla Adapter | 19.45% | 11.36% |
| CLIP QRNN Adapter | 23.70% | 15.35% |
| LaViLa Vanilla Adapter | 19.48% | 12.28% |
| LaViLa QRNN Adapter | 29.12% | 17.10% |
| EgoVLP Vanilla Adapter | 18.09% | 10.89% |
| EgoVLP ST Adapter | 17.36% | 10.20% |
| EgoVLP QRNN Adapter | 28.75% | 16.50% |
| EgoVLP RetNet Adapter | 25.69% | 13.67% |

Table 4: **Streaming Recall Results for two temporal tolerances**. We provide additional results to those in Table 2 for one minute clips when evaluating smaller window tolerances.

streaming queries phase (Section 4) we keep the earliest of the duplicated annotations, which results in a bias towards *early starts*. Second, some latency is expected from the models, as the event start might not be immediately obvious.

We emphasize that different values can be used, and include additional results comparing a tighter tolerance window ($[-2, +5]$) to the values reported in Table 4. Note that the same trends discussed in Section 6 hold, albeit with smaller values, as fewer predictions are considered correct.

## C  Model Architecture Details

### C.1  Vision Transformers (ViT)

Vision Transformers (ViT) adapt the transformer architecture to process 2D images. Given an image denoted $I \in \mathbb{R}^{H \times W \times C}$ where $H$ and $W$ are the spatial dimensions, and $C$ is the channel dimension, ViT extracts $N$ non-overlapping image patches, $x_i \in \mathbb{R}^{h \times w}$, performs a linear projection to $d$ dimensions, and then embeds them into a sequence of 1D tokens $\mathbb{R}^{w \times h \times d}$. Specifically, the sequence of tokens input to the transformer encoder is represented as:

$$\mathbf{z} = [z_{cls}, \mathbf{E}x_{i_{hw}}] + \mathbf{p}, \tag{3}$$

where $z_{cls}$ is an optional class embedding, $\mathbf{E}x_{i_{hw}}$ denotes the linear projection of image patches into $d$ dimensions, $x_{i_{hw}} \to z_{i_d}$, and the index notation $i_{hw}$ is reshaped into $(hw)d$ using Einstein notation to indicate the transformation from 2D spatial dimensions to a flattened sequence. The learned positional embedding $\mathbf{p} \in \mathbb{R}^{N \times d}$ is added to these tokens to retain positional information.

The tokens $\mathbf{z}$ are passed through an encoder consisting of a sequence of $L$ transformer layers, each performing the operations:

$$\begin{aligned} \mathbf{y}^{\ell} &= MHSA(LN(\mathbf{z}^{\ell})) + \mathbf{z}^{\ell} \\ \mathbf{z}^{\ell+1} &= MLP(LN(\mathbf{y}^{\ell})) + \mathbf{y}^{\ell} \end{aligned} \tag{4}$$

where MHSA denotes Multi-Headed Self-Attention, LN is layer normalization, and MLP consists of two linear projections with a GELU non-linearity between them. The token-dimensionality $d$ remains fixed throughout all layers.

### C.2  Image to Video Adaptation

Given a pre-trained image model $\mathcal{F}$ and a set of videos, with each video $\mathbf{V} \in \mathbb{R}^{T \times H \times W \times C}$, and where $T$ represents time, $H$ and $W$ are the spatial dimensions of an image or frame, and $C$ is the channel dimension, our objective is to bridge the modality gap between the image level pretraining of the model, and the spatio-temporal downstream video task. We aim to adapt the image model into a spatio-temporal video model $\mathcal{F}* : \mathbb{R}^{T \times H \times W \times C} \to \mathbb{R}^{n_t \times n_h \times n_w \times d}$ while reusing as many parameters from $\mathcal{F}$ as possible.

The modified Equation 4 employed in Spatio-Temporal Transformer Blocks is as follows:

$$\begin{aligned}
\mathbf{y}_{temp}^{\ell} &= TA(\mathbf{z}^{\ell}) \\
\mathbf{y}^{\ell} &= MHSA_{spatial}(LN(\mathbf{y}_{temp}^{\ell})) + \mathbf{z}^{\ell} \\
\mathbf{z}^{\ell+1} &= MLP(LN(\mathbf{y}^{\ell})) + \mathbf{y}^{\ell}
\end{aligned} \tag{5}$$

Importantly, because the spatial layers operate independently on every frame, $MHSA_{spatial}$ requires that we fold the temporal dimension of input tokens into the batch dimension and that the patches are flattened, *i.e.* $\mathbf{y}_{temp}^{\ell}$ is reshaped to $\mathbb{R}^{n_t \times n_h \cdot n_w \times d}$. Here we assume the leading dimension would correspond to the aforementioned "batch dimension".

For models with a shallower architecture, like the ViT-B, an additional temporal adapter module is inserted after the spatial attention layer to enhance temporal processing capabilities.

### C.3 ST-Adapter

A natural first choice of architecture for temporal aggregation $TA$ is to use 1D Convolutions as in ST-Adapter [54].

In ST-Adapter convolutions are applied in parallel across time steps, mixing the information from neighbouring timesteps, which constitutes the primary temporal aggregation operation. Figure 5 shows an example of how temporal information is aggregated across timesteps for each patch-wise tubelet with shape $\mathbb{R}^{n_t \times 1 \times 1 \times d}$. This operation can be performed efficiently in PyTorch by folding the patch dimensions $n_h, n_w$ of input tokens to the temporal aggregation model into the batch dimension such that $\mathbf{z}$ is reshaped to $\mathbb{R}^{n_h \cdot n_w \times n_t \times d}$. As in Equation 5 the leading dimension is the batch dimension.

As mentioned in the main text, and following prior work [54, 56] our implementation of ST-Adapter operates on a reduced dimension $d' < d$ to reduce computational cost. Given an input sequence $\mathbf{z}^{\ell} \in \mathbb{R}^{n_h \cdot n_w \times n_t \times d}$ we first downsample each patch from $d$ to $d'$. We then convolve the resulting sequence to obtain an output sequence $\mathbf{o} \in \mathbb{R}^{n_h \cdot n_w \times n_t \times d'}$.

$$\mathbf{o} = \mathbf{W}_s * \mathbf{z}^{\ell} \tag{6}$$

where $\mathbf{W}_s \in \mathbb{R}^{k \times d' \times d'}$ is a convolutional filter bank with kernel size $k$, and $*$ denotes a masked convolution along the time dimension.

The final step in our adapter is the upsampling of the sequence back up from $d'$ to $d$.

It is important to note that the use of a convolution operation requires that we pad our input sequence $\mathbf{z}^{\ell}$ appropriately to avoid temporally downsampling the sequence. To produce an equal sized output, we apply zero-padding around its boundaries. Specifically, for a convolution kernel of size $k$, we pad the sequence with a total of $k - 1$ zeros. However, it is crucial to note that the padding does not need to be symmetric on either side of the sequence. Instead, we adjust the value of $k$ along with the proportion of the padding on the left and right of the sequence to enable our model to consider additional context from frames from the future or past when generating $f_t$ and $s_t$. We refer to the amount of future frames considered as the lookahead window, and conversely, to the amount of past frames considered as the lookback window.

An important limitation of convolutional approaches is the limited temporal context, as the kernel defines only a fixed window of inputs that it can process at any given time, restricting the range of dependencies that can be captured within that window. Although stacking multiple convolutional layers can extend this window —where the receptive field $M$ layers with kernel width $k$ is calculated as $k + (M - 1) \times (k - 1)$— this still imposes a constraint on the breadth of temporal dependencies that can be effectively captured compared to recurrent approaches. For instance, in our experiments where we with CLIP-base [52] where we add a single 1D convolution adapter per block the effective receptive field is 12 frames (*i.e.* a maximum of 12 seconds of prior context can be considered by the model).

### C.3.1 Initializing ST-Adapter

Following work from existing adapter modules across NLP [50, 51] and Video [54, 56], we initialize our adapter such that it initially approximates the identity function for better training dynamics. To

this end we zero initialize the linear upsampling layer following the adapter resulting in the overall adapter which carries over the value received by its residual connection, thereby approximating the identity function. The motivation for this is to slowly integrate the adapter module into the larger image backbone.

## C.4   QR-Adapter

The choice of architecture for the temporal aggregation layer $TA$ fundamentally governs the dynamics of the model scaling with respect to sequence length (*i.e.* video duration). However, most existing approaches either rely on base architectures with fixed receptive fields, such as 1D Convolutions in ST-Adapter [54], which makes them unsuitable for modelling long sequences.

In QR-Adapter use QRNN which alternates convolutions -applied in parallel across time steps- with minimalist recurrent temporal pooling layers without any trainable parameters. As was the case for ST-Adapter, QRNN aggregates temporal information across timesteps for each patch-wise tubelet with shape $\mathbb{R}^{n_t \times 1 \times 1 \times d}$ as shown in Figure 5. As with ST-Adapter, this operation can be performed efficiently by folding the patch dimensions $n_h, n_w$ of input tokens to the temporal aggregation model into the batch dimension, such that $\mathbf{z}$ is reshaped to $\mathbb{R}^{n_h \cdot n_w \times n_t \times d}$. Again, the leading dimension is the batch dimension.

Our QR-Adapter also operates on a reduced dimension $d' < d$ to reduce computational cost. Given an input sequence $\mathbf{z}^\ell \in \mathbb{R}^{n_h \cdot n_w \times n_t \times d}$ we first downsample each patch from $d$ to $d'$. We then apply two separate convolutions to the resulting sequence to obtain a candidate state vector $\mathbf{s} \in \mathbb{R}^{n_h \cdot n_w \times n_t \times d'}$, and a forget-vector $\mathbf{f}$ of the same shape:

$$
\begin{aligned}
\mathbf{s} &= \tanh(\mathbf{W}_s * \mathbf{z}^\ell) \\
\mathbf{f} &= \sigma(\mathbf{W}_f * \mathbf{z}^\ell)
\end{aligned}
\tag{7}
$$

where $\mathbf{W}_z$, and $\mathbf{W}_f$, each in $\mathbb{R}^{k \times d' \times d'}$, are depth-wise convolutional filter banks with kernel size $k$, and $*$ denotes a masked convolution along the time dimension. Finally, the output sequence $\mathbf{h} \in \mathbb{R}^{n_h \cdot n_w \times n_t \times d'}$ is computed by a recurrent pooling of each individual timestep's representation in time:

$$
\mathbf{h}_t = \mathbf{f}_t \odot \mathbf{h}_{t-1} + (1 - \mathbf{f}_t) \odot \mathbf{s}_t,
\tag{8}
$$

where $\mathbf{s}_t$ and $\mathbf{f}_t$ denote the state and forget vectors corresponding to timestep $t$ from the convolution operations, and $\mathbf{h}_{t-1}$ denotes the output of the recurrent pooling function from the prior timestep. The recurrent formulation allows for running the model indefinitely, as processing a new frame only requires that we save the most recent token $\mathbf{h}_{n_t}$ of the recurrent memory, and use it to initialize the hidden state of the recurrent pooling layer for future iterations, as visualized in Figure 5b.

The final step in our adapter is the upsampling of the sequence back up from $d'$ to $d$, resulting in the output sequence $\mathbf{h}^\ell \in \mathbb{R}^{n_h \cdot n_w \times n_t \times d}$.

It is important to note that the use of a convolution operation in the QRNN again requires that we pad our input sequence $\mathbf{z}^\ell$ appropriately. The definitions of lookahead and lookback window from the previous section carry over to our implementation of QR-Adapter.

### C.4.1   Initializing QR-Adapter

As in the previous section, we initialize our adapter such that it initially approximates the identity function by zero initializing the linear upsampling layer following the QRNN. Similarly, we also initialize the forget gate convolution filter bank $\mathbf{W}_f$ to be all zero, and initialize the forget gate convolution's bias to a fixed negative value $(-5)$ such that the forget gate output $\mathbf{f}_t$ is approximately zero. This leads to an initial recurrent pooling update of $\mathbf{h}_t = \mathbf{s}_t$, thus removing the impact of the prior hidden state. This setup allows for the steady integration of temporal modelling into the preexisting image model in a principled manner.

## C.5   RN-Adapter

Each block in Retentive Networks (RetNet) [65] resembles a Transformer block [66], but replaces the Self Attention operation for *Retention*. As with Self Attention each token is projected into a query,

key, and value vector which are combined by comparing queries and keys, and scaling values by said score:

$$Q = (XW_Q) \odot \Theta, \quad K = (XW_K) \odot \overline{\Theta}, \quad V = XW_V$$
$$\text{Retention}(X) = (QK^\intercal \odot D)V \tag{9}$$

where $\Theta_n = e^{in\theta}$, $\overline{\Theta}$ is the complex conjugate of $\Theta$, and $D \in \mathbb{R}^{|x| \times |x|}$ combines causal masking and exponential decay. For further reading refer to the Retention Networks paper [65].

Importantly, said formulation (Equation 9) foregoes any non-linear operations and therefore can be rewritten as a Recurrent Neural Network:

$$S_n = \gamma S_{n-1} + K_n^\intercal V_n$$
$$\text{Retention}(X_n) = Q_n S_n, \quad n = 1, \cdots, |x| \tag{10}$$

where $Q, K, V$ are the same as in Equation 9, and $\gamma$ is the base that is exponentially decayed in $D$.

We take advantage of both formulations in our RN-Adapter. During training we leverage the parallel representation of Retention to train with GPUs efficiently, while during inference we can instead use the recurrent formulation to avoid repeating computation and minimize memory use. Indeed, the Transformer's attention scores and $D$ scale quadratically with sequence length, making it unfeasible to use these models for some of our video sequences (which can exceed 1.5 hours or 5400 frames long).

As with ST-Adapter and QR-Adapter, we use Retention to aggregate temporal information across timesteps for each patch-wise tubelet. Again, the adapter operates on a reduced dimension $d' < d$ to reduce computational cost, which is later upsampled back up from $d'$ to $d$.

### C.5.1 Initializing RN-Adapter

As in previous sections where we described initalization for ST-Adapter and QR-Adapter, we initialize our RN-Adapter by initially setting the weights in the linear upsampling layer to zero.

## D Implementation Details and Hyperparameters

### D.1 Event Start Detection

In Section 3 we broadly define the models used to obtain the event start scores from the streaming context. Briefly, the query and the right-aligned representations of the visual features up to the current time-step $i$ are used to compute the score for said time-step. In practice, during training we consider a context of at most $w_s - 1$ past frames in addition to the current one, and make predictions for each of these:

$$M^{(i)}(\{f_{i-w_s+1}, \ldots, f_i\}) \mapsto \{s^{(i)}_{i-w_s+1}, \ldots, s^{(i)}_i\}. \tag{11}$$

such that we can provide denser supervision to the model. However, for inference we discard the scores for past frames and only consider the score given to the most recent:

$$s_i = s^{(i)}_i. \tag{12}$$

We train all models with windows of $w_s = 60$ frames at one frame per second. Exceptionally, we train RetNet variants with fewer frames $w_s = 30$ as we found the model training to be unstable with longer sequence lengths. This instability is likely due to the large exponents in computations like $\gamma^{(w_s)}$ (where $0 < \gamma < 1$), which can cause numerical instability like underflow, where these values can approach zero too closely, destabilizing the computations and model training. Setting $w_s = 30$ during training stabilizes calculations and improves model training reliability.

As per Section 5.2 during training we sample windows of training we report validation metrics on randomly sampled windows of $w_s$ frames. The threshold for a score to be considered a prediction is selected from a set of 20 uniformly spaced candidate values ranging between the minimum and the maximum observed probabilities (obtained by $\sigma(s_i)$ where $\sigma$ is the sigmoid function):

$$\text{ts} = \text{np.linspace}(\text{probs.min}(), \text{probs.max}(), 20)$$

| Model | $w_s$ During Training | Link to Experiment |
|---|---|---|
| ZS CLIP ViT-B/16 | - | link |
| CLIP Vanilla Adapter | 60.00 | link |
| CLIP QRNN Adapter | 60.00 | link |
| LaViLa Vanilla Adapter | 60.00 | link |
| LaViLa QRNN Adapter | 60.00 | link |
| EgoVLP ST Adapter | 60.00 | link |
| EgoVLP Vanilla Adapter | 60.00 | link |
| EgoVLP QRNN Adapter | 60.00 | link |
| EgoVLP RetNet Adapter | 30.00 | link |
| EgoVideo Vanilla Adapter | 50.00 | link |

Table 5: **Experiment links**. We include Weights and Biases links (wandb.ai) to the logs for every experiment in Table 2 and Figure 6.

We report each metric at every threshold for randomly sampled windows of $w_s = 60$ and $w_s = 300$, and select the threshold that maximizes the SR@1 with 5 seconds of allowable anticipation and 10 seconds of latency for that video duration. For testing, we evaluate the performance on the same window sizes, $w_s = 60$ and $w_s = 300$, but apply the thresholds established through the randomly sampled windows during training on a standardized validation set. This approach ensures that instead of using randomly sampled windows, we employ a consistent set of windows for evaluating all models. We include the standardized validation set with our data release. We additionally evaluate performance on the full videos, which does not require sampling windows (as all frames are considered).

## D.2  Model Hyperparameters

To ensure that the adapters are comparable we adjust the downsampling dimension $d'$ such that the number of parameters in each adapter is roughly equal to the number of parameters in our ST-Adapter. See Table 3 to verify parameter counts for each one. For ST-Adapter we carry over the value used in [54] (*i.e.* $d' = \frac{d}{2}$).

Where we can, we set hyperparameters to be the same across all models trained. Other than the number of frames considered during training we also adjust the learning rate, reducing it when encountering training instabilities. Table 5 includes links to the logs of all referenced experiments, and includes the exact command, hyperparameters, and seed to replicate the results. The same links include information as to what compute we used and training time. Experiments run on private servers between one to two days at a carbon cost of approximately $27gCO_2$ equivalent per hour (estimated using mlco2.github.io [67] and historical information taken from electricitymaps for the state of California).

## E  Additional Dataset Generation Details

As noted in Section 4, we translate two existing sources of temporal annotations for Ego4D [3] and the annotations from EgoExoLearn [60] into our own streaming queries. For this we need to prompt the generative language model to generate reasonable and quality generations. To this end we combine the temporal annotation with dense video captions to contextualize and ground the temporal annotation in the contents of the video.

The query generation process is integral to our system, designed to construct contextually aware queries from existing temporally grounded language annotations. This section delineates the steps involved in filtering the data, generating the scripts, and synthesizing the queries, ensuring relevance and specificity.

### E.1  Ego4D Moments Annotations

**Data Filtering.** The initial step involves rigorously filtering the narrations and moments data to ensure consistency and completeness. Narrations are retained only if they have a 'complete' status,

while moments data is conditioned on the availability of corresponding narrations. Furthermore, to prevent redundancy and enhance computational efficiency, we first sort annotations temporally, and then employ a duplication check that cross-references new annotations against previously generated queries, thereby omitting any repeated data processing.

**Script Generation.** For each video annotation, a *script* is dynamically generated to serve as the context for the query generation model. We provide one example output per script, but vary the example among generations to encourage diversity. To facilitate this, we select the example query based on a deterministic hash function amalgamating various identifiers (video, clip, and annotator IDs). This approach ensures consistent example selection across different executions. We also prepare a second *disambiguated* example for scenarios where an event has occurred before, which aids in minimizing ambiguity in the queries.

The script comprehensively outlines the event itself, embedding detailed timings and associated (prior) narrations. It includes both high-level summaries and specific event narrations, thereby enriching the context provided to the model. This detailed script is designed to highlight the event's temporal occurrence within the video and contextualize with the rest of the narrations.

**Query Synthesis.** Our query synthesis pipeline leverages OpenAI's API to process the prepared script and generate a query that aligns with the specified context. The system formulates the input for the API by encapsulating the system prompt along with the user-generated script as context.

The model is tasked with producing a query that is not only relevant but also specific to the particular instance of the event, especially in cases where the event has previously occurred. This specificity is crucial for applications requiring precise action based on the video content. Should the model generate a query that fails to meet the criteria of relevance or specificity (e.g., due to ambiguous or incomplete responses), the system employs a retry mechanism. This mechanism adjusts the inputs based on the error encountered and re-invokes the model, striving to refine the query output.

We enforce that the model generate the following intermediate outputs in order: a boolean indicating whether the event has occurred before, a detailed event label, a specific query tailored to the event's context, an answer string that directly corresponds to the query prompt, and a boolean indicating whether the query is specific only to this instance of the event. Generating these intermediate outputs in a specific order is crucial for grounding each subsequent output, ensuring the overall coherence and specificity of the generated queries.

The complete prompt is used to condition a Large Language Model to generate the requested questions and answer candidates in a JSON format. The chosen language model is GPT-4. We set the sampling temperature to zero and decode greedily (for replicability). We include the exact prompts used here: first the system prompt, and then the user prompt showcasing an example script (trimmed at ". . ." to fit the page):

---

**Query Generation System Prompt**

You're helping me generate a new dataset for an online assistant that receives a first person view of the world (via a head mounted camera, e.g., augmented reality glasses). In short, will receive an EVENT and must convert it into a query in which a human asks an assistant to identify the point at which the event becomes true (the start of the event).

```
Eg. {
    "query": "Let me know when it's safe to cross the street.",
    "ans": "You may cross the street now."
}
```

You won't have access to the full video, instead I'll provide narrations of the events in the video, and high level summaries. Use them to enrich the query with context (e.g., say "talk to the cashier" or "interact with person X" instead of "converse/interact with someone"), but remember to keep the query grounded in the video (*do not* invent details). Importantly, the narrations might contain mistakes, especially with the language.

The videos are in first person and #C ALWAYS refers to the camera wearer, i.e., the person that is using the assistant. #O refers to others in the video.

If the event B has already occurred before, make sure that the query is unambiguous. For example, by referring to another event A that happened before.

---

You should only return a JSON file with a single query formatted as indicated (with the keys in the same order):

```
{
    "event_has_occurred_before": true|false,
    "event": string,
    "request": string,
    "query": string,
    "ans": string,
    "query_is_specific_only_to_this_event": true|false
}
```

## Video Narration Script

**Event you should generate a "start" query for:** use_phone (0:05:47 - 0:05:49)
**VIDEO NARRATIONS:** The high level descriptions of the video are:

- (0:00:00 - 0:05:00) #Summary C used the phone, watered the flowers, walked outside the house compound then arranged the documents.
- (0:04:30 - 0:05:49) #Summary C walked upstairs with the documents, cleared the working station then used the phone

The low level events in the video are:

- (0:00:02) #C C uses the phone
- (0:00:09) #C C gets up
- (0:00:09) #C C takes a cup of water
- (0:00:12) #C C opens door curtain
- . . .
- (0:03:00) #C C uses the phone
- (0:03:05) #C C opens television door stand
- (0:03:09) #C C takes out some documents
- (0:03:21) #C C puts earphones to the cabinet
- (0:03:25) #C C puts in some of the documents to the cabinet
- . . .
- (0:04:25) #C C gets up
- (0:04:30) #C C puts down a water bottle
- (0:04:34) #C C arranges the documents
- (0:04:38) #C C takes a water bottle
- (0:04:46) #C C walks upstairs
- (0:05:17) #C C puts documents on top of the work table
- (0:05:21) #C C puts down the water bottle
- (0:05:23) #C C takes out the tin lid from the table
- (0:05:26) #C C puts tin lid on the other side of the table
- (0:05:31) #C C takes water bottle
- (0:05:33) #C C puts the water bottle aside
- (0:05:36) #C C removes dust from the table
- (0:05:47) **EVENT STARTS HERE**
- (0:05:48) #C C uses phone
- (0:05:49) **EVENT ENDS HERE**

Remember to format the query in first person, as if the user is asking the assistant. (like the examples in the prompt).
**Sample Query Configuration:**

```
{
    "event_has_occurred_before": false,
    "event": "enter_/_exit_building",
    "request": "reminder to check mail.",
    "query": "Ask me to check the mail when I leave the house.",
    "ans": "Remember to check the mail.",
    "query_is_specific_only_to_this_event": true
}
```

Again, the event you should generate the query for is use_phone (occurs between 0:05:47 - 0:05:49) Remember that query should be a reminder to do something when the EVENT STARTS (event starts to occur). Remember to make the query unambiguous if the event has already occurred before (if 'event_has_occurred_before: true'). If the query is not specific **only** to the given instance of the event, then set query_is_specific_only_to_this_event: false.

## E.2 Ego4D NLQ (Natural Language Query) annotations

**Data Filtering.** Similar to the Moment annotations pipeline, narrations are filtered to ensure they are marked 'complete', ensuring data quality and consistency. Unlike the general query process, the NLQ pipeline includes specific filtering based on template and word lists. Annotations with templates not in the whitelist or containing blacklisted words are excluded from further processing to maintain query quality and relevance. We specifically filter events that are likely to be ungrounded or those that refer to spoken interactions between people. We again check for duplicates by looking for significant temporal overlap between new annotations and previously processed ones to avoid generating queries for duplicate events. Annotations with high overlap are marked and skipped, ensuring each query is unique.

**Script Generation.** We again select an example from a pre-specified list based on a hash of unique identifiers (video, clip, annotation IDs) to ensures consistency across different runs. For events that are detected to have occurred previously, a disambiguated example is also prepared to encourage reduced ambiguity in the generated queries.

The script itself remains unchanged.

**Query Synthesis.**

While this part mostly is the same as for Moments annotations, a few substantial differences exist. Most important is that we also request that the LLM identifies whether the event is grounded in narrations. This is necessary because many of the NLQ annotations refer to background events that do not show up in the Narrations. If we kept these annotations it would be hard to identify whether the event has occurred before. By filtering NLQ annotations that we are not sure we can detect previous instances of we ensure that the following generations are accurate. Particularly $event\_has\_occurred\_before$ and $query\_is\_specific\_only\_to\_this\_instance\_of\_event$.

---

Query Generation System Prompt NLQ

You're helping me generate a new dataset for an online assistant that receives a first person view of the world (via a head mounted camera, e.g., augmented reality glasses). In short, it will receive a question and must convert it into a query in which a human asks an assistant to identify the point at which the event starts. For example:

```
{
    "query": "Let me know when it's safe to cross the street.",
    "alert": "You may cross the street now."
}
```

You won't have access to the full video; instead, I'll provide narrations of the events in the video and high-level summaries. Use them to enrich the query with context (e.g., say "talk to

---

the cashier" or "interact with person X" instead of "converse/interact with someone"), but remember to keep the query grounded in the video (*do not* invent details). Importantly, the narrations might contain mistakes, especially with the language.
The videos are in first person and #C ALWAYS refers to the camera wearer, i.e., the person using the assistant. #O refers to others in the video.
If event B has already occurred before, make sure that the query is unambiguous, for example, by referring to another event A that happened before.
You should only return a JSON file with a single query formatted as indicated (with the keys in the same order):

```json
{
    "question": string,
    "event": string,
    "event_is_grounded_in_narrations": true|false,
    "event_has_occurred_before": true|false,
    "request": string,
    "query": string,
    "alert": string,
    "query_is_specific_only_to_this_instance_of_event": true|false
}
```

### E.3  EgoExoLearn (Natural Language Query) annotations

Similar to the Ego4D dataset, EgoExoLearn contains dense captions in text, describing the main occurrences in the video. We filter annotations to keep only the egocentric videos. We do not filter any based on content using word blacklists. An important difference with Ego4D annotations is that each of the dense captions is annotated with both start and end times. This is significant in two ways. First, it means we can leverage the dense captions both for contextualizing and as the base annotation that we adapt into our EgoSDQES annotation. As was the case with both sources of annotations from Ego4D we only include the contextualizing events that occur prior to the event we're generating a query for. Other than these changes the LLM prompt remains mostly unchanged with respect to the NLQ annotation pipeline from the previous section.

## F  Broader Impacts and Limitations

Our work presents the first benchmarks for encouraging model development towards, and measuring progress of, a critical task for time-sensitive detection of events described with natural language queries. This is a task which has potential for positive impact to settings where low-latency video understanding can improve safety. Below, we discuss the potential for other broader impacts and limitations.

**Dataset Bias and Deployment.** Our proposed benchmarks build directly on the data and the annotations collected in prior work, Ego4D [3]. This means that our benchmark inherits limitations of these datasets, with respect to their data distributions (e.g., skewed representation of gender, geographical origin, culture, among others). This means that performance of models trained for SDQES on our generated dataset EgoSDQES may have their performance dependent on such factors, as has been observed in other literature for vision and language models [68]. Our techniques tackle problems related to event understanding that may involve people, and as such, could be misused in ways that threaten people's privacy if deployed without taking appropriate precaution.

**Other Limitations.** In addition to the above, we note some technical limitations with the construction of our streaming datasets. First, the adaptation of temporal localization annotations into event start detection ones, and the inherent ambiguities with language descriptions, means that the annotated event boundary may not be frame-perfect. We aim to mitigate the impacts of these by introduction of an acceptable start window in our main streaming recall metric (see Section B.3), but this is something that can continue to be a focus for future work. In addition, as discussed in the main paper, the construction of our dataset relies on often faulty pre-existing annotations, as well as the capacity of Large Language Models to follow the precise instruction and understand the context provided.

Any failure in this process can result in low-quality generations, which also affects models trained on the data.

# G EgoSDQES Datasheet

## G.1 Motivation

**For what purpose was the dataset created?** Was there a specific task in mind? Was there a specific gap that needed to be filled? Please provide a description.

EgoSDQES was created for the Streaming Detection of Queried Event Start (SDQES) task. This task aims to identify the start of complex events as described by a natural language query with high accuracy and low latency. The dataset facilitates the development and evaluation of models capable of multimodal understanding in a streaming video context, particularly for applications requiring quick reaction times like robotics and augmented reality.

**Who created this dataset (e.g., which team, research group) and on behalf of which entity (e.g., company, institution, organization)?**

The dataset is a product of the SVL group within Stanford Computer Science, Stanford University.

## G.2 Composition

**What do the instances that comprise the dataset represent (e.g., documents, photos, people, countries)?** Are there multiple types of instances (e.g., movies, users, and ratings; people and interactions between them; nodes and edges)? Please provide a description.

The instances in the dataset represent streaming video frames annotated with natural language queries. These queries describe specific events that start within these video frames. Each query is accompanied by start and end timestamps and a response string.

**How many instances are there in total (of each type, if appropriate)?**

EgoSDQES has a total of 12767 query annotations spanning 1773 distinct videos.

**Does the dataset contain all possible instances or is it a sample (not necessarily random) of instances from a larger set?** If the dataset is a sample, then what is the larger set? Is the sample representative of the larger set (e.g., geographic coverage)? If so, please describe how this representativeness was validated/verified. If it is not representative of the larger set, please describe why not (e.g., to cover a more diverse range of instances, because instances were withheld or unavailable).

The dataset is a sample of instances from a larger set of egocentric video data. It is not necessarily representative of all possible scenarios but is curated to include a diverse range of activities, viewpoints, and camera movements. The representativeness for specific tasks is validated through its design to challenge models with real-world scenarios encountered in egocentric video understanding.

**What data does each instance consist of? "Raw" data (e.g., unprocessed text or images) or features?** In either case, please provide a description.

Each instance consists of raw video in *.mp4* format coupled with temporally and contextually relevant natural language queries. Each query is accompanied by start and end timestamps in seconds encoded as floating point numbers and a response string.

**Is there a label or target associated with each instance?** If so, please provide a description.

Each query is associated with a start and end timestamp which specifies the slice of the video in which the queried event is occurring. As it pertains to SDQES, only the start timestamp is relevant. However, we use both the start and end timestamps for training. See Section 5.2.

**Is any information missing from individual instances?** If so, please provide a description, explaining why this information is missing (e.g. because it was unavailable). This does not include intentionally removed information but might include, e.g., redacted text.

N/A

**Are relationships between individual instances made explicit (e.g., users' movie ratings, social network links)?** If so, please describe how these relationships are made explicit.

Relationships between instances (e.g., sequential video frames and their corresponding queries) are made explicit (as they share a common $video\_uid$). The dataset is structured to allow models to use prior video frames and language cues to predict events, emphasizing temporal and contextual relationships necessary for streaming applications.

**Are there recommended data splits (e.g., training, development/validation, testing)?** If so, please provide a description of these splits, explaining the rationale behind them.

EgoSDQES includes recommended splits for training and validation. These splits are designed to help in the systematic training and evaluation of models, ensuring that they can generalize across different scenarios presented in the dataset. We divide individual videos and all their associated annotations into one of $\{train, val\}$ such that there is no overlap between training and validation instances. This split is done according to the official Ego4D splits [3] to avoid data contamination when evaluating models trained on Ego4D.

**Are there any errors, sources of noise, or redundancies in the dataset?** If so, please provide a description.

Like any real-world dataset, especially one involving egocentric videos, the videos contain some level of noise. Additionally, mistakes in existing annotations in the source dataset percolate and can be amplified by our pipeline. Finally, when an event starts is sometimes ambiguous. For example, the event "opening the refrigerator" can be thought to start when the hand touches the refrigerator handle, or later when the refrigerator door begins to open.

**Is the dataset self-contained, or does it link to or otherwise rely on external resources (e.g., websites, tweets, other datasets)?** If it links to or relies on external resources, a) are there guarantees that they will exist, and remain constant, over time; b) are there official archival versions of the complete dataset (i.e., including the external resources as they existed at the time the dataset was created); c) are there any restrictions (e.g., licenses, fees) associated with any of the external resources that might apply to a future user? Please provide descriptions of all external resources and any restrictions associated with them, as well as links or other access points, as appropriate.

The dataset does not heavily rely on external resources beyond the initial video content, which was released under the Ego4D license. EgoSDQES is self-contained with respect to the primary task of detecting queried events in video streams. The annotations generated for EgoSDQES will be made available to the public under the MIT license, permitting the use of the text data for research and commercial applications.

**Does the dataset contain data that might be considered confidential (e.g., data that is protected by legal privilege or by doctor-patient confidentiality, data that includes the content of individuals non-public communications)?** If so, please provide a description.

 No.

**Does the dataset contain data that, if viewed directly, might be offensive, insulting, threatening, or might otherwise cause anxiety?** If so, please describe why.

No

**Does the dataset relate to people?** If not, you may skip the remaining questions in this section.

The use of egocentric video could potentially include personal or sensitive content. People recorded in the videos consented to the Ego4D license. However, the creators of Ego4D implemented a variety of de-identification techniques, focusing mainly on maintaining a controlled setting where all participants provided informed consent. When videos are recorded in public areas an effort was made to keep personally identifiable details obscured. Our contribution on top of those annotations does not require additional consent or de-identification efforts.

**Does the dataset identify any subpopulations (e.g., by age, gender)?** If so, please describe how these subpopulations are identified and provide a description of their respective distributions within the dataset.

No.

**Is it possible to identify individuals (i.e., one or more natural persons), either directly or indirectly (i.e., in combination with other data) from the dataset?** If so, please describe how.

No.

**Does the dataset contain data that might be considered sensitive in any way (e.g., data that reveals racial or ethnic origins, sexual orientations, religious beliefs, political opinions or union memberships, or locations; financial or health data; biometric or genetic data; forms of government identification, such as social security numbers; criminal history)?** If so, please provide a description.

No.

### G.3   Collection Process

**How was the data associated with each instance acquired?** Was the data directly observable (e.g., raw text, movie ratings), reported by subjects (e.g., survey responses), or indirectly inferred/derived from other data (e.g., part-of-speech tags, model-based guesses for age or language)? If data was reported by subjects or indirectly inferred/derived from other data, was the data validated/verified? If so, please describe how.

The video data (directly observable) was acquired from the Ego4D dataset. The generated queries (directly observable text) were generated using Large Language Models, specifically GPT4. These LLMs take as input visual narrations and an event description (both also directly observable text) and generate a new "streaming" query.

**What mechanisms or procedures were used to collect the data (e.g., hardware apparatus or sensor, manual human curation, software program, software API)?** How were these mechanisms or procedures validated?

The video and narration data were downloaded from the official Ego4D website https://ego4d-data.org. Text data were generated using API access to GPT-4 via OpenAI. Additional details on generation and curation are available in the main paper, Sections 4 and E.

**If the dataset is a sample from a larger set, what was the sampling strategy (e.g., deterministic, probabilistic with specific sampling probabilities)?**

We consider all videos in Ego4D that have been annotated with both Narrations and one of Moments or NLQ temporal annotations. Queries are generated for the annotations that pass the filters detailed in Sections 4 and E.

**Who was involved in the data collection process (e.g., students, crowdworkers, contractors) and how were they compensated (e.g., how much were crowdworkers paid)?**

The annotation process was automated. Additional verification of the quality of generations was carried out by authors of the paper.

**Over what timeframe was the data collected? Does this timeframe match the creation timeframe of the data associated with the instances (e.g., recent crawl of old news articles)?** If not, please describe the timeframe in which the data associated with the instances was created.

The original videos in the Ego4D dataset were collected between 2019 and 2021. Our own annotation efforts were carried out in the first half of 2024.

**Were any ethical review processes conducted (e.g., by an institutional review board)?** If so, please provide a description of these review processes, including the outcomes, as well as a link or other access point to any supporting documentation.

No

**Does the dataset relate to people?** If not, you may skip the remaining questions in this section.

Yes

**Did you collect the data from the individuals in question directly, or obtain it via third parties or other sources (e.g., websites)?**

The video and narration data were collected following the Ego4D data access guidelines, with the necessary consent from participants: https://ego4d-data.org/docs/start-here/.

**Were the individuals in question notified about the data collection?** If so, please describe (or show with screenshots or other information) how notice was provided, and provide a link or other access point to, or otherwise reproduce, the exact language of the notification itself.

The Ego4d paper outlined privacy and ethical safeguards, including informed consent from camera wearers and de-identification of personal data. Details on specific instructions to participants are not provided. The privacy statement can be accessed at https://ego4d-data.org/pdfs/Ego4D-Privacy-and-ethics-consortium-statement.pdf

**Did the individuals in question consent to the collection and use of their data?** If so, please describe (or show with screenshots or other information) how consent was requested and provided, and provide a link or other access point to, or otherwise reproduce, the exact language to which the individuals consented.

The Ego4d paper details privacy measures, such as obtaining informed consent from camera wearers. Specific instructions to participants are not disclosed. Refer to the Ego4D privacy statement for more information.

**If consent was obtained, were the consenting individuals provided with a mechanism to revoke their consent in the future or for certain uses?** If so, please provide a description, as well as a link or other access point to the mechanism (if appropriate).

The Ego4d paper details privacy practices, such as permitting camera users to modify their video footage. The privacy statement is available at https://ego4d-data.org/pdfs/Ego4D-Privacy-and-ethics-consortium-statement.pdf.

**Has an analysis of the potential impact of the dataset and its use on data subjects (e.g., a data protection impact analysis) been conducted?** If so, please provide a description of this analysis, including the outcomes, as well as a link or other access point to any supporting documentation.

We direct readers to the Ego4D paper for detailed discussion on the impact of the video dataset. Ego4D has implemented multiple privacy measures, such as depersonalizing sensitive data and anonymizing visuals, to mitigate privacy risks.

### G.4  Preprocessing/cleaning/labeling

**Was any preprocessing/cleaning/labeling of the data done (e.g., discretization or bucketing, tokenization, part-of-speech tagging, SIFT feature extraction, removal of instances, processing of missing values)?** If so, please provide a description. If not, you may skip the remainder of the questions in this section.

We do not modify the videos provided by Ego4D in any way. We use automated tools to curate our generated queries. Details are available in the main paper, Sections 4 and E.

**Was the "raw" data saved in addition to the preprocessed/cleaned/labeled data (e.g., to support unanticipated future uses)?** If so, please provide a link or other access point to the "raw" data.

We include JSON files with all the generations, including redundant, non-specific and non-grounded ones along with the main dataset release. These can be found in the linked GitHub repository, or

in files within the sdqesdataset.github.io/dataset/intermediate_generations/ directory (*e.g.* unfiltered moments validation data).

**Is the software used to preprocess/clean/label the instances available?** If so, please provide a link or other access point.

Yes. All code for generation, filtering etc. is provided in the supplementary materials, as well as at github.com/sdqesdataset/sdqes_generation.

### G.5    Distribution

**Will the dataset be distributed to third parties outside of the entity (e.g., company, institution, organization) on behalf of which the dataset was created?** If so, please provide a description.

The dataset will be made publicly available and can be used for both research and commercial purposes under the Ego4D license.

**How will the dataset be distributed (e.g., tarball on website, API, GitHub)** Does the dataset have a digital object identifier (DOI)?

The dataset will be distributed as a JSON file describing the unique identifier for each clip, the associated question, the five answer options, the label, and additional clip information that facilitates the tracing of the clip back to the original Ego4D data, such as the Ego4D video identification of the clip's source video, among other details. In addition, download tools to acquire and pre-process the video RGB data will also be provided on our website.

**When will the dataset be distributed?**

The full dataset will be made available upon the acceptance of the paper before the camera-ready deadline.

**Will the dataset be distributed under a copyright or other intellectual property (IP) license, and/or under applicable terms of use (ToU)?** If so, please describe this license and/or ToU, and provide a link or other access point to, or otherwise reproduce, any relevant licensing terms or ToU, as well as any fees associated with these restrictions.

EgoSDQES be publicly released under the MIT license, which allows direct public use of the video and text data for both research and commercial purposes.

**Have any third parties imposed IP-based or other restrictions on the data associated with the instances?** If so, please describe these restrictions, and provide a link or other access point to, or otherwise reproduce, any relevant licensing terms, as well as any fees associated with these restrictions.

No

**Do any export controls or other regulatory restrictions apply to the dataset or to individual instances?** If so, please describe these restrictions, and provide a link or other access point to, or otherwise reproduce, any supporting documentation.

No

### G.6    Maintenance

**Who will be supporting/hosting/maintaining the dataset?**

The authors of the paper will support maintaining the dataset.

**How can the owner/curator/manager of the dataset be contacted (e.g., email address)?**

The website includes a reference to the official EgoSDQES dataset email: sdqesdataset@gmail.com.

**Is there an erratum?** If so, please provide a link or other access point.

We will publish errata on the Github repository.

**Will the dataset be updated (e.g., to correct labeling errors, add new instances, delete instances)?** If so, please describe how often, by whom, and how updates will be communicated to users (e.g., mailing list, GitHub)?

Yes, we plan to improve the quality of the generations by including additional curation steps. Future versions of the dataset will be posted to the official dataset website at sdqesdataset.github.io.

**If the dataset relates to people, are there applicable limits on the retention of the data associated with the instances (e.g., were individuals in question told that their data would be retained for a fixed period of time and then deleted)?** If so, please describe these limits and explain how they will be enforced.

No.

**Will older versions of the dataset continue to be supported/hosted/maintained?** If so, please describe how. If not, please describe how its obsolescence will be communicated to users.

Yes. We will keep old versions of the data alongside the new versions.

**If others want to extend/augment/build on/contribute to the dataset, is there a mechanism for them to do so?** If so, please provide a description. Will these contributions be validated/verified? If so, please describe how. If not, why not? Is there a process for communicating/distributing these contributions to other users? If so, please provide a description.

We open-source our query generation pipeline to facilitate future efforts to extend the dataset. If contributors want to make changes to the official data or code bases they can do so by submitting a Pull Request on the appropriate GitHub repository. The authors commit to verifying the contribution. Changes made will only affect future versions of the dataset and will be communicated accordingly.

