# OpenReview forum: "Streaming Detection of Queried Event Start"
_NeurIPS.cc/2024/Datasets_and_Benchmarks_Track — NeurIPS 2024 Track Datasets and Benchmarks Poster_

### Official Review · Reviewer_TeQt · 2024-07-23
**Review Comments**

**Rating:** 7
**Confidence:** 4
**Correctness:** Yes.
**Clarity:** Yes.

**Review:**

The paper introduces a groundbreaking task of streaming event detection with significant implications for real-time applications, though it faces limitations in dataset biases and generalizability considerations that should be addressed for broader impact.
Pros:
1. Introduces a new and challenging task that is highly relevant to real-world applications.
2. Provides a comprehensive benchmark and metrics for evaluating performance in the SDQES task.
Cons:
1. The reliance on existing datasets (Ego4D) may inherit biases and limitations from the source data.
2. The complexity of the task may limit the generalizability of the findings to other types of video content.
3. While the adapter-based approach is efficient, it may not capture all nuances of spatio-temporal relationships in video data.

**Strengths:**

The introduction of the Streaming Detection of Queried Event Start (SDQES) task is a significant contribution that addresses a current gap in the field of video understanding, particularly for real-time applications. The work is highly relevant to the broader research community, including researchers in robotics, autonomous driving, augmented reality, and embodied computer vision, who need to detect events with low latency. The research is methodologically sound, with a clear problem statement, well-defined objectives, and a thorough experimental evaluation.

**Additional Feedback:**

None.

**Documentation:**

Yes.

**Ethics:**

No.

**Limitations:**

Yes.

**Opportunities For Improvement:**

1. The reliance on the Ego4D dataset could introduce a selection bias, potentially limiting the generalizability of the findings to other types of video data or event categories.
2. The adapter-based approach, while efficient, might not capture the full complexity of spatio-temporal dynamics in video data compared to more complex models.

**Relation To Prior Work:**

Yes.

**Summary And Contributions:**

The paper introduces a groundbreaking task of streaming event detection with significant implications for real-time applications, though it faces limitations in dataset biases and generalizability considerations that should be addressed for broader impact.

The submission presents a novel task and approach within the domain of video understanding, titled "Streaming Detection of Queried Event Start" (SDQES). The task is designed to address the need for quick reaction to user-defined events in real-time, as seen in applications like robotics, autonomous driving, and augmented reality. The core contributions of the paper are as follows:
1. The authors introduce the SDQES task, which involves identifying the onset of a complex event described by a natural language query within a streaming video, emphasizing high accuracy and low latency.
2. A new benchmark, EgoSDQES, is presented, leveraging the Ego4D dataset. This benchmark is tailored to study the streaming multimodal detection of diverse events in an egocentric video setting.
3. The authors describe an innovative data generation pipeline using Large Language Models (LLMs) for event extraction and query generation, ensuring the relevance and specificity of the annotations for the task.

---

> ### Author Rebuttal · Authors · 2024-08-17
>
> **R3: The reliance on the Ego4D dataset could introduce a selection bias, potentially limiting the generalizability of the findings to other types of video data or event categories.**
>
> Thank you for your feedback. In designing the dataset we strove for diversity in both the video sources and the source annotations that we adapt for EgoSDQES, which is why we chose Ego4D as our starting point. Ego4D offers great diversity, with 931 participants across 74 locations in 9 countries, providing a broad spectrum of unscripted footage collected "in the wild" and annotated in natural language.
>
> That being said, we chose to limit the domain of the dataset to egocentric videos to ensure the relevancy and applicability of the dataset to real-world scenarios. By focusing exclusively on egocentric videos, we harness the unique challenges and dynamics present in these types of visual data, such as variable lighting, rapid camera movement, and frequent occlusions.
> We emphasize that the same data generation pipeline and proposed model architecture can be repurposed to generate and solve an allocentric version of the SDQES task.
>
> ---
>
> **R3: The adapter-based approach, while efficient, might not capture the full complexity of spatio-temporal dynamics in video data compared to more complex models.**
>
> Indeed, we agree that a more powerful spatio-temporal model could likely outperform our choice of adapted-models. However, our main motivation when proposing the baseline models was to support the extremely long sequences present in the EgoSDQES with moderate amounts of computational power. Table 3 of the supplementary material supports this, as it shows that even small-windowed self-attention methods are extremely computationally expensive compared to the proposed approaches. We think future research can leverage our work to explore these avenues.

---

> > ### Author Response · Authors · 2024-08-27
> > **Follow-Up on Feedback and Rating**
> >
> > Dear Reviewer TeQt, thank you for your earlier feedback. We believe we have addressed your concerns through our response. As the discussion period is ending soon, we wanted to check if you have any additional comments. If our responses have adequately addressed your concerns, we kindly invite you to consider increasing your ratings. We look forward to your feedback.

---

### Official Review · Reviewer_E6vj · 2024-07-25

**Rating:** 6
**Confidence:** 3
**Correctness:** It appears that there are no concerns…
**Clarity:** It appears that there are no concerns…

**Review:**

- The task discussed in the paper appears to be original and challenging for the model to recognize, and it seems to have potential contributions to specific industries.
- The clarity and articulation of the proposal are also pronounced.

**Strengths:**

The proposal of a new benchmark and dataset called SDQES timely addresses key challenges and establishes an appropriate baseline model through a combination of relevant technologies.
The task's industrial contributions are aligned with the adoption and expansion of an egocentric video dataset, which seems to have laid the groundwork for discussion in the proposed benchmark.

**Additional Feedback:**

none

**Documentation:**

The proposed method discusses the composition and organization of the dataset. However, it does not appear to address maintenance or expansion.

**Ethics:**

It appears that there are no concerns regarding ethics.

**Limitations:**

Moment retrieval critically requires learning the alignment between natural language and video representations, and robust text-video understanding is needed to detect moments corresponding to reference texts in online streaming data. This underscores the importance of dataset scale for learning strong associations between natural language and video. Due to insufficient datasets, models may have to rely on backbones that possess strong language comprehension and video understanding capabilities. The construction of an integrated understanding necessary to achieve the proposed benchmark might be inadequate with a limited dataset.

**Opportunities For Improvement:**

none

**Relation To Prior Work:**

The paper somewhat addresses how it differentiates from prior research in the problem domain it seeks to address.

**Summary And Contributions:**

- The proposed paper introduces a benchmark for predicting the start of events in online settings through text queries.
- The proposed task maintains complex challenges while differentiating itself from prior benchmarks, and is considered to have both academic and industrial value.

---

> ### Author Rebuttal · Authors · 2024-08-17
>
> **R2: Moment Retrieval necessitates large-scale data to learn video-language alignment. Models may have to rely on backbones that possess strong language comprehension and video understanding capabilities.**
>
> We agree with this characterization of the complexities of the SDQES task.
> The fact that events are described by open-vocabulary natural language results in an increased challenge for video understanding models. In our proposed baselines, we address the need for large scale video-language training by leveraging diverse pretraining sources, namely, CLIP based models use CLIP pretraining, EgoVLP based models are pre-trained on the entirety of Ego4D, and LaViLA based models are trained with both.
> While we agree that at its current scale our dataset EgoSDQES is insufficient to completely learn the task without additional pre-training sources, we emphasize that this is true of most modern benchmarks.

---

> > ### Author Response · Authors · 2024-08-27
> > **Follow-Up on Feedback and Rating**
> >
> > Dear Reviewer E6vj, thank you for your earlier feedback. We believe we have addressed your concerns through our response. As the discussion period is ending soon, we wanted to check if you have any additional comments. If our responses have adequately addressed your concerns, we kindly invite you to consider increasing your ratings. We look forward to your feedback.

---

### Official Review · Reviewer_h4vt · 2024-07-25
**A new benchmark for streaming detection of queried event start**

**Rating:** 8
**Confidence:** 5
**Correctness:** Yes
**Clarity:** Yes

**Review:**

This paper presents a significant contribution to the community by offering a novel task and baseline method that could advance real-time event detection systems.

Pros:
1. This work tackles streaming detection of queried event start, which is a novel and important problem in the field of multimodal video understanding. I believe this is very beneficial for next-generation AR systems.
2. Propose a practical solution using adapter modules for efficient online learning.
3. This work provides a comprehensive experimental evaluation with various models and adapters.

Cons:
1. All the videos are sampled from Ego4D. While Ego4D already has good demographic diversity, it is good to include video sources from multiple datasets to cover better diversity in demographic and camera settings. Some datasets contain crowd-sourced egocentric videos and also support converting to the SDQES format, for example, HoloAssist and EgoExoLearn.
2. It would be better to test non-LLM specialized models, like online action detection models, online action segmentation models.

**Strengths:**

1. This work tackles streaming detection of queried event start, which is a novel and important problem in the field of multimodal video understanding. I believe this is very beneficial for next-generation AR systems.
2. Propose a practical solution using adapter modules for efficient online learning.
3. This work provides a comprehensive experimental evaluation with various models and adapters.

**Additional Feedback:**

N/A

**Documentation:**

Yes

**Limitations:**

Yes

**Opportunities For Improvement:**

1. All the videos are sampled from Ego4D. While Ego4D already has good demographic diversity, it is good to include video sources from multiple datasets to cover better diversity in demographic and camera settings. Some datasets contain crowd-sourced egocentric videos and also support converting to the SDQES format, for example, HoloAssist and EgoExoLearn.
2. It would be better to test non-LLM specialized models, like online action detection models, online action segmentation models.

**Relation To Prior Work:**

Yes

**Summary And Contributions:**

This paper proposes an interesting and useful benchmark aiming for streaming detection of queried event start, instantiated using egocentric videos. New task-specific metrics to study streaming multimodal detection of diverse events are also introduced. A simple adapter-based baseline approach is used  in the experiments, while multiple combinations of VLM and adapters are evaluated.

---

> ### Author Rebuttal · Authors · 2024-08-17
>
> **R1: Additional datasets sources other than Ego4D would help cover better diversity in demographic and camera settings.**
>
> Thank you for your insightful comments. We acknowledge the importance of diversity in demographic and camera settings and have carefully considered other egocentric datasets in our pipeline. We directly inherit the distribution from Ego4D. As for the inclusion of other sources, note that our pipeline requires 1) diverse temporal annotations, and 2) human-provided narrations (i.e., descriptive dense video captions) to temporally disambiguate actions effectively. Among other temporally annotated egocentric datasets, we evaluated several, but they unfortunately have limitations for our purpose:
>
> 1) **HoloAssist**: While it includes annotations that the authors term narrations, these are video-level summaries. Additionally, the temporal annotations are not in natural language but are instead limited to noun-verb classes, which limits its generalizability \[[1](https://arxiv.org/abs/2309.17024)\].
>
> 2) **EgoExoLearn**: The low-level hand verb-noun temporal annotations (also termed narrations) included do not align with the natural language dense captions we require for our pipeline \[[2](https://arxiv.org/abs/2403.16182)\].
>
> 3) **H2O, HOI4D, Meccano, Assembly101**: These datasets do not provide narrations in the form required for our task, have few action classes for the temporal annotations (often verb-noun classes), and are restricted to lab environments \[[4](https://arxiv.org/abs/2104.11181)\]\[[5](https://arxiv.org/abs/2203.01577)\]\[[6](https://arxiv.org/abs/2010.05654)\]\[[7](https://arxiv.org/abs/2203.14712)\].
>
> 4) **EpicKitchens**: Among the other datasets, EpicKitchens is the only one that includes dense captions similar to Ego4D. However, its temporal annotations are limited to valid combinations of a few verbs and nouns, and videos are restricted to kitchen settings \[[3](https://arxiv.org/abs/1804.02748)\].
>
> The datasets that have annotations in formats that do not align with our requirements could result in issues like repeated labels and a subsequent reduction in data diversity.
> Furthermore, the restricted environments of some datasets would dilute the variety of scenarios covered, further diminishing the dataset’s overall diversity.
> Most critically, the high-quality natural language narrations provided by Ego4D that are essential for disambiguating actions are absent in some of the datasets.
> Given these evaluations, we determined that including any of these other datasets to our pipeline would add biases and potentially reduce the generalizability of models trained on the resulting data.
>
> ---
>
> **R1: It would be better to test other non-LLM specialized models like online action detection models, online action segmentation models.**
>
> We clarify that the  models included in our evaluation are not based on LLMs in the way current SoTA video language models are, where the core functionality relies on a massive language model with billions of parameters \[[8](https://arxiv.org/abs/2305.06988),[9](https://arxiv.org/abs/2407.00634),[10](https://arxiv.org/abs/2408.03326)\]. Instead, our approach uses a smaller language Transformer (e.g. CLIP language encoder is 63 million parameters) solely for the text encoding. The video encoding is kept entirely separate in a different backbone. Both modalities don’t mix until the very last layer of the model where we compute the cosine similarity of the video and text embeddings.
>
> Regarding testing other models: Existing models designed for ODAS and other online video tasks lack the text component necessary to understand the language query that describes the event in SDQES. Note that unlike traditional ODAS, we do not have pre-defined action categories, and instead the query is in free-form natural language.
> The most similar task to our work is the new (and concurrent to our work) Streaming Dense Video Captioning task \[[11](https://arxiv.org/abs/2404.01297)\] . However, the released model architecture 1) doesn't allow for discriminative tasks, and 2) the weights weren’t trained on egocentric videos.
>
> We also explored adapting an existing ODAS model, OADTR \[[12](https://arxiv.org/abs/2106.11149)\], to better fit the requirements of our task.
> Specifically, we introduced a language encoder into the OADTR architecture. This modification involved using the language encoders already employed in our other baselines to generate language embeddings. These embeddings then replaced the "task token" in OADTR, allowing the model to interpret free-form natural language queries alongside visual content.
> We have included the comparative results of this adapted model in the revised results table as an additional baseline.
> However, it's important to note that OADTR operates in a windowed mode, which limits its ability to maintain context outside of the current window, potentially affecting its performance on tasks requiring broader temporal understanding.
>
> We hope that this work will enable future research to adapt advances in ODAS and other real time video tasks to SDQES.
>
>
> | Method | SR@1↑ | SR@2↑ | SR@3↑ |
> |-|-|-|-|
> | Zero-Shot CLIP | 7.9 | 11.6 | 14.0 |
> | | | | |
> | CLIP + OADTR | 7.7 | 10.5 | 12.8 |
> | CLIP + Adapter | 8.9 | 13.7 | 17.2 |
> | CLIP + QR-Adapter | 9.1 | 14.1 | 18.7 |
> | | | | |
> | EgoVLP + OADTR | 8.4 | 13.4 | 17.4 |
> | EgoVLP + Adapter | 8.4 | 13.0 | 16.7 |
> | EgoVLP + QR-Adapter | 9.7 | 14.1 | 17.9 |

---

> > ### Author Response · Authors · 2024-08-27
> > **Follow-Up on Feedback and Rating**
> >
> > Dear Reviewer h4vt, thank you for your earlier feedback. We believe we have addressed your comments through our response and additional experiments. As the discussion period is ending soon, we wanted to ask if you have any further comments. If our responses have sufficiently addressed your concerns, we kindly invite you to consider increasing your ratings. We look forward to your feedback.

---

> > > ### Comment · Reviewer_h4vt · 2024-08-28
> > > **Thank you for your comments**
> > >
> > > As can be seen from my rating, I believe this is one of the good papers to be accepted.
> > >
> > > However, the rebuttal in fact does not fully address my concerns.
> > >
> > > Firstly, as I understand, the EgoExoLearn dataset contains exactly the natural language captions (**not** verb and nouns as said in the rebuttal), which is even more suitable than Ego4D since it is annotated with temporal boundaries but not narration points.
> > >
> > > Secondly, I believe the authors did not answer my questions. SoTA video language models also do not use LLMs. What I am asking is to adapt SoTA models for offline NLQ, into this online setting. By this means we can fully see the effectiveness of the proposed adapter scheme.

---

> > > > ### Author Response · Authors · 2024-08-29
> > > > **Thank you for your feedback**
> > > >
> > > > **R1: Additional dataset sources other than Ego4D would help cover better diversity in demographic and camera settings.**
> > > >
> > > > Thank you for your continued feedback and insights. We appreciate your suggestion to consider the EgoExoLearn dataset. We looked into the dataset again, and realized that the earlier stages of the EgoExoLearn pipeline do include natural language annotations, which we initially overlooked due to the final output focusing on verb-noun pairs. **We will extend our current dataset**, using these intermediate annotations to improve its diversity and robustness. We will release this improved version of our dataset by the end of this Friday and will update experiments in the following days.
> > > >
> > > > ---
> > > >
> > > > **R1: Adaptation of SoTA models for offline NLQ into the online setting would be more effective to demonstrate the effectiveness of the proposed adapter scheme.**
> > > >
> > > > We realize there could be two possible interpretations of the comment in your review on testing online models: 1) directly “test non-LLM specialized models, like online action detection models, online action segmentation models”, or 2) adapting SoTA offline NLQ models into the online setting. We would like to address both to ensure clarity and completeness.
> > > > 1. If the question is about directly testing online action/segmentation models, then as we explained above, we modified an existing online action detection model (OADTR) to fit our streaming detection task. We reported the results of this adaptation in our rebuttal to demonstrate how the model performs in the context of our study.
> > > >
> > > > 2. If the question is about adapting SoTA offline NLQ Models to an online streaming context, then we actually have done what you suggested in the original paper. Specifically, we included the EgoVLP and LAVILA backbones in our experiments: EgoVLP was the state-of-the-art offline model on EgoNLQ until recently, and LAVILA is the state-of-the-art offline dual encoder model. If there are any other offline NLQ models that you’d like to see adapted to our online setting, please don’t hesitate to let us know!
> > > >
> > > > Thank you! We hope this clarifies your concerns. We will follow up once we have the updated dataset by Friday; meanwhile, please don’t hesitate to let us know if any concerns remain.

---

> ### Comment · Reviewer_h4vt · 2024-08-30
> **Thank you for your response**
>
> I believe EgoVLP and LAVILA are not SoTA offline NLQ models. I think not only the encoder+adapter fashion should be in the experiment, but also specialized models. For example, [GroundNLQ](https://arxiv.org/pdf/2306.15255) and [NaQ](https://openaccess.thecvf.com/content/CVPR2023/papers/Ramakrishnan_NaQ_Leveraging_Narrations_As_Queries_To_Supervise_Episodic_Memory_CVPR_2023_paper.pdf) uses EgoVLP as feature extractor and adds a specialized model for this task, and the result is much better than EgoVLP itself. [GroundVQA](https://github.com/Becomebright/GroundVQA) is another recent work that can also be easily applied to NLQ.

---

> > ### Author Response · Authors · 2024-08-31
> > **Thank you**
> >
> > Thank you.  We initially did not include them as they are extensions of EgoVLP with specialized components for the NLQ task, but indeed, having them would strengthen the work.  As you suggested, we are currently training an adapted version of [EgoVideo](https://arxiv.org/abs/2406.18070) which is based on GroundNLQ and represents the current SoTA for EgoNLQ tasks. We will include these results in the updated paper.  If we can get these numbers by the discussion deadline, we will also post an update here.
> >
> > Additionally, as promised, the extended version of our dataset is now available [here](https://sdqesdataset.github.io/dataset/all_egoexo.csv). We will update our experiments on this new data accordingly in the coming days.
> >
> > Thank you again for your valuable suggestions.

---

> > > ### Author Response · Authors · 2024-09-01
> > > **Thank you for your feedback**
> > >
> > > Thank you once again for your insightful suggestions. We are pleased to provide an update on the new experiments incorporating the specialized models you recommended, and the most recent state of the art backbone, [EgoVideo](https://arxiv.org/abs/2406.18070).
> > >
> > > We have evaluated two main variations:
> > > 1. **EgoVideo+Adapters**: We added our adapters to the backbone.
> > > 2. **EgoVideo+GroundNLQ\***: We modified the NLQ version of EgoVideo, which includes the adapted backbone and the Multi-modal Multi-scale Transformer Encoder from GroundNLQ, to output a binary classification score indicating the presence of the queried action in the most recent frame.
> > >
> > > **Current results are promising and have already advanced the state-of-the-art in our dataset**.
> > > For live results, including all experiments, hyperparameters, plots and notes, please visit our [experiment report](https://wandb.ai/erictang000/sdqes/reports/EgoVideo--Vmlldzo5MjE0MDQx).
> > > Given the tight deadline, we expect that additional improvements could be made (e.g., with further hyperparameter tuning or extended training time).
> > >
> > > We sincerely appreciate your time and effort and hope these updates, along with the extension we made to the dataset, address your concerns.

---

### Author Rebuttal · Authors · 2024-08-17

We thank the reviewers for their valuable and insightful feedback. We appreciate the reviewers' recognition of the novelty and importance of our proposed SDQES task. Our work is acknowledged as introducing a **groundbreaking** (R3-TeQt) and **novel** (R1-h4vt, R2-E6vj) problem in multimodal video understanding, with **significant implications for real-time applications** like robotics, autonomous driving, augmented reality, and specific industries (R1-h4vt, R2-E6vj, R3-TeQt). We are pleased that the reviewers found our practical solution using adapter modules to be **efficient** (R1-h4vt, R3-TeQt) and that the **comprehensive nature of our benchmark and evaluation** was noted (R1-h4vt, R3-TeQt). Furthermore, we value the recognition of our task's potential contributions to both academia and industry (R2-E6vj, R3-TeQt), and the recognition of the **innovative approach of our data generation pipeline** (R3-TeQt).

As requested by Reviewer 1 (R1-h4vt), we have explored how to adapt existing ODAS models to better align with the requirements of our task. Specifically, we have modified the OADTR architecture by introducing a language encoder. Additionally, we have addressed several comments and questions regarding the suitability and sufficiency of the Ego4D dataset for our task, as well as the limitations of an adapter-based architecture.

We have provided detailed responses to each question below, and invite the reviewers to consider increasing their ratings if our answers sufficiently address their concerns.

Thank you.

---

### Decision · Program_Chairs · 2024-09-26

**Decision:**

Accept (Poster)

**Comment:**

This paper proposes the task of streaming detection of when an event begins, with a query of open vocabulary text, with low latency. All three reviewers gave this paper positive scores (6,7,8), appreciative of the novel and challenging task, as well as the provided baseline approach. There are clear applications for models with such capabilities, including robotics, autonomous driving, and augmented reality.

Two reviewers expressed concern about the limitations of the source videos being drawn solely from Ego4D. This concern is somewhat mitigated by Ego4D’s geographic/demographic diversity and scale, as the authors point out; if the authors are interested in more data with similar characteristics, then Ego4D’s successor Ego-Exo4D may also be a good source to draw from. On the other hand, both datasets involve humans with head-mounted cameras performing activities; it may be a stretch that such a dataset will generalize to robotics or autonomous driving, which is used by the paper as motivation. Regardless, even if focused on human activity, this dataset and benchmark should still serve as an important contribution for advancing video understanding. The AC recommends acceptance.